# A non-stationary Markov model for economic evaluation of grass pollen allergoid immunotherapy

**Massimo Bilancia**[1]*, **Giuseppe Pasculli**[2], **Danilo Di Bona**[3]

**1** Ionic Department in Legal and Economic System of Mediterranean (DJSGEM), University of Bari Aldo Moro, Taranto, Italy, **2** Department of Computer, Control, and Management Engineering Antonio Ruberti (DIAG), La Sapienza University, Rome, Italy, **3** School and Chair of Allergology and Clinical Immunology, Department of Emergency and Organ Transplantation (DETO), University of Bari Aldo Moro, Bari, Italy

\* massimo.bilancia@uniba.it

**Data Availability Statement:** All relevant data are within the manuscript and its Supporting Information files.

**Funding:** The authors received no specific funding for this work.

## Abstract

### Introduction

Allergic rhino-conjunctivitis (ARC) is an IgE-mediated disease that occurs after exposure to indoor or outdoor allergens, or to non-specific triggers. Effective treatment options for seasonal ARC are available, but the economic aspects and burden of these therapies are not of secondary importance, also considered that the prevalence of ARC has been estimated at 23% in Europe. For these reasons, we propose a novel flexible cost-effectiveness analysis (CEA) model, intended to provide healthcare professionals and policymakers with useful information aimed at cost-effective interventions for grass-pollen induced allergic rhino-conjunctivitis (ARC).

### Methods

Treatments compared are: 1. `no AIT`, first-line symptomatic drug-therapy with no allergoid immunotherapy (AIT). 2. `SCIT`, subcutaneous immunotherapy. 3. `SLIT`, sublingual immunotherapy. The proposed model is a non-stationary Markovian model, that is flexible enough to reflect those treatment-related problems often encountered in real-life and clinical practice, but that cannot be adequately represented in randomized clinical trials (RCTs). At the same time, we described in detail all the structural elements of the model as well as its input parameters, in order to minimize any issue of transparency and facilitate the reproducibility and circulation of the results among researchers.

### Results

Using the `no AIT` strategy as a comparator, and the Incremental Cost Effectiveness Ratio (ICER) as a statistic to summarize the cost-effectiveness of a health care intervention, we could conclude that:

- `SCIT` systematically outperforms `SLIT`, except when a full societal perspective is considered. For example, for $T = 9$ and a pollen season of 60 days, we have ICER = €16,729 for `SCIT` vs. ICER = €15,116 for `SLIT` (in the full societal perspective).

**Competing interests:** The authors have declared that no competing interests exist.

- For longer pollen seasons or longer follow-up duration the ICER decreases, because each patient experiences a greater clinical benefit over a larger time span, and Quality-adjusted Life Year (QALYs) gained per cycle increase accordingly.

- Assuming that no clinical benefit is achieved after premature discontinuation, and that at least three years of immunotherapy are required to improve clinical manifestations and perceiving a better quality of life, ICERs become far greater than €30,000.

- If the immunotherapy is effective only at the peak of the pollen season, the relative ICERs rise sharply. For example, in the scenario where no clinical benefit is present after premature discontinuation of immunotherapy, we have ICER = €74,770 for `SCIT` vs. ICER = €152,110 for `SLIT`.

- The distance between `SCIT` and `SLIT` strongly depends on under which model the interventions are meta-analyzed.

## Conclusions

Even though there is a considerable evidence that `SCIT` outperforms `SLIT`, we could not state that both `SCIT` and `SLIT` (or only one of these two) can be considered cost-effective for ARC, as a reliable threshold value for cost-effectiveness set by national regulatory agencies for pharmaceutical products is missing. Moreover, the impact of model input parameters uncertainty on the reliability of our conclusions needs to be investigated further.

## Introduction

Allergic rhino-conjunctivitis (ARC) is an IgE-mediated disease that occurs after exposure to indoor or outdoor allergens, or to non-specific triggers such as smoke and viral infections [1, 2]. Symptoms include rhinorrhea, nasal obstruction, itchy nose, and repetitive sneezing, accompanied by ocular symptoms, such as itchy, red, watery, and swollen eyes [3]. Symptoms of ARC are also classified as seasonal or perennial on the basis of their temporal pattern. Seasonal allergies result from exposure to airborne substances that appear only during certain times of the year, with grass pollen being by far the most common cause of disease [4]. Depending on whether the symptoms last less or more than 4 days per week or 4 weeks per year, ARC frequency can be divided respectively into intermittent and mild. ARC severity instead, can be classified as mild or more severe respectively when symptoms are present but they do or do not interfere with overall quality of life, exacerbate co-existing asthma, induce sleep disturbances or impair daily activities and school/work performances [5]. Furthermore, ARC and asthma are frequently co-occurring conditions, and rhinitis typically precedes the onset of asthma [6]. Indeed, it has been estimated that between 19% and 38% of patients with allergic rhinitis suffer from concomitant asthma worldwide [7].

Treatment options for seasonal ARC include environmental control of allergens, pharmacological therapies for symptom control, and allergen immunotherapy [4]. For the initial treatment of moderate to severe seasonal ARC in persons aged 12 years or older, a combination of an intranasal corticosteroid plus an oral/intranasal antihistamine is recommended [8, 9]. Allergen immunotherapy (AIT) should be considered for patients suffering from moderate or severe persistent symptoms which interfere with usual daily activities or sleep, despite

compliant and appropriate drug therapies, or in patients experiencing unacceptable side-effects associated with symptomatic first-line treatments [2, 5]. AIT may also be considered in less severe ARC where a patient wishes to take advantage of its potential for preventing disease progression towards asthma [1]. There is a growing body of evidence supporting that AIT induces favorable immunological changes, defined as the persistence of clinical benefits for at least 1-year after treatment discontinuation [10]. Another possible role of the latter treatment in children is reducing the risk of new allergen sensitization [11], although the evidence still remain poor [12, 13].

With regard to AIT, subcutaneous injection (SCIT) has been the predominant method of administration. However, over the last two decades sublingual application of allergens (SLIT) has increased, and it is now the dominant approach in several European countries [14, 15]. The SCIT regimen comprises an initial loading phase, consisting of a weekly administration of allergen extracts for 1-3 months, followed by monthly maintenance injections [16, 17]. On the contrary, in SLIT regimen the build-up period is not needed and patients receive a once-daily fixed-dose, which is administered continuously throughout the year or pre/co-seasonally, depending on the allergen that triggers the symptoms [18]. Maintenance doses for both SCIT and SLIT have traditionally been recommended to be continued for at least 3 years [19, 20].

A benefit from both SCIT and SLIT compared with placebo has been consistently demonstrated in randomized controlled trials (RCTs), but the extent of this effectiveness in terms of clinical benefit is still unclear [21–23]. Indirect comparisons of SCIT with SLIT were suggestive of SCIT being more beneficial, but no statistically significant difference was found between these two interventions using a combined symptom-medication score [22]. Moreover, discrepancies between the clinical settings of RCTs and those of real clinical practice make the assumptions of these modeling studies uncertain and limit the validity of their conclusions.

In addition, the economic aspects and burden of these therapies are not of secondary importance. The prevalence of ARC has been estimated at 23% in Europe [24, 25], and its cost for the national health services has followed an exponential growth. For example, using data from the National Medical Expenditure Survey it was estimated by [26] that total cost of the ARC condition in the USA, in 1994 Dollars, was $1.23 billion (95% confidence interval, $846 million to $1.62 billion), with direct medical expenses accounting for 94% of total costs. Less outdated evidence was reported in [27], a study based on a probabilistic prevalence-based cost of illness model, where it was estimated that the total economic burden in Italy associated with respiratory allergies and their main co-morbidities was €7.33 billion in 2015 Euros (95% CI: €5.99 – €8.82), with €5.32 billion for direct medical costs). In the same study, the authors estimated that the annual average direct cost for symptomatic ARC therapy per patient were €129 (Min: €89 – Max: €168) versus €320 (Min: €258 – Max: €381) and €365 (Min: €303 – Max: €427) for SLIT and SCIT, respectively.

Hence, these data suggest that it would be inappropriate to ignore the weight of the economic burden deriving from the use of allergoid immunotherapy for seasonal ARC. Cost-effectiveness analysis (CEA) summarizes the problem of valuing health related outcomes by estimating an incremental cost due to treatment per unit change in outcome [28, 29]. The latter combines information about morbidity and quality of life, in order to generate a quality-adjusted life year (QALY) value. A year in perfect health is associated with 1 QALY and death with 0 QALY, whereas other possible states are ranked between these two extremes [30].

The CEA studies reviewed in [23] found both SCIT and SLIT being more beneficial than symptomatic treatments in terms of incremental cost-effectiveness ratios (ICERs). From their results, an additional expense of about €22,000 was necessary for a unit increment in QALY over the patients follow-up window. However, [23] pointed out there were issues around transparency and robustness of input parameters for most studies. Moreover, none of the

cost–utility analyses were conducted by independent researchers. Similarly, [31] claimed that the quality of most previously conducted studies is poor, due to the general lack of attention in characterizing uncertainty and handling real-life situations.

For all of the above reasons, we developed a non-stationary Markovian model for CEA of allergoid immunotherapy. This model was flexible enough to reflect those treatment-related problems often encountered in real-life, but that cannot be adequately represented in RCTs [32]. At the same time, we described in detail all the structural elements of the model as well as its input parameters, in order to minimize any issue of transparency and facilitate the reproducibility and the circulation of the results. Our model is intended to provide healthcare professionals and policymakers with useful information for cost-effective interventions, which might inform decision-making on selecting the option with the best value.

The emphasis of the paper will be to focus on the simulation of selected scenarios, each based on a specification representing alternative assumptions in order to match relevant clinical situations. In the final section of this manuscript we explored further techniques to quantify parameter estimates uncertainty such as probabilistic sensitivity analysis (PSA), a Monte Carlo method to simulate the sampling distribution of the joint mean cost and efficacy.

## Materials and methods

From the description above we can define three strategies for each treatment option for seasonal ARC. Each strategy comprises a certain number of health states:

1. `no AIT`, in which symptomatic drug therapy is administered, comprising the three following states:

   - `No Asthma (NA_1)`, including patients with ARC without asthma.

   - `Asthma (A_1)`, including patients with ARC and asthma.

   - `Death (D_1)`.

2. `SCIT`, comprising five health states:

   - `AIT + No Asthma (ANA_2)`, including patients with ARC treated with `SCIT`.

   - `AIT + Asthma (AA_2)`, including patients with ARC treated with `SCIT`, who develop or have developed co-existing asthma.

   - `No AIT + no Asthma (NANA_2)`, including patients that prematurely abandoned `SCIT` or have completed the 3-year course of treatment.

   - `No AIT + Asthma (NAA_2)`, including patients that prematurely abandoned `SCIT` or have completed the 3-year course of treatment, who develop or have developed co-existing asthma.

   - `Death (D_2)`.

3. `SLIT`, comprising five health states

   - `AIT + No Asthma (ANA_3)`, similar to `ANA_2` once the necessary changes have been made.

   - `AIT + Asthma (AA_3)`, as above.

   - `No AIT + no Asthma (NANA_3)`, as above.

- No AIT + Asthma $(NAA_3)$, as above.

- Death $(D_3)$.

For each strategy, we simulate the evolution of a cohort of $N = 1000$ individuals in discrete time. The number of individuals in a given health state at the end of Markov cycle $t$ is denoted as $n_t(s)$, with $s \in \mathcal{S}_i$, where $\mathcal{S}_i$ is the set of health states of strategy $i \in \{1, 2, 3\}$ (for example $n_t(AA_3)$), with the constraint that:

$$N = \sum_{s \in \mathcal{S}_i} n_t(s)$$

The vector giving the probability of being in a given state at the end of Markov cycle $t$ will be denoted as $\pi_{i,t}$ (for strategy $i$). In particular, $\pi_{i,0}$ is the vector of being in a given state at the end of the first period $t = 0$. If not otherwise stated, $t$ indicates the end of the $t$-th Markov cycle.

## Patient population

Similarly to the stationary Markov model developed by Verheggen et al. [33], patients included in the simulated cohort had the following characteristics (common to all three strategies):

- Suffered from moderate-to-severe grass-pollen seasonal acute ARC. At entry, none of the patients suffered from co-existing chronic asthma. However, mild-to-moderate allergic asthma can develop as the simulation runs.

- Positive grass allergen-specific skin prick test and/or elevated serum grass pollen specific IgE.

- Average age of 35.9 years at entry, reflecting the median of the mean age of patients in the adult studies included in the meta-analysis by [23].

## Model structure

A Markov model was considered to simulate, for each strategy, the probability distribution of being in each health state over a discrete sequence of time periods. We assumed the duration of each time period (or Markovian cycle) to be of one year. The beginning of therapy was in $t = 0$, with a follow-up of $T$ years, with $T = 9$ in our base case. Although this is not mandatory, a longer follow-up might be clinically justifiable. Subsequent periods were numbered 1, 2, . . .,9, for a total of 10 years. For each of the three strategies, the relationships of the patients of the cohort with the available treatments were the following:

- no AIT: symptomatic drug treatment including antihistamine (desloratadine 5mg/die) and nasal corticosteroid (budesonide nasal spray 200 mcg/die; two 50 mcg puffs twice a day). Symptomatic medications were taken daily during grass pollen season from the first year and were continued over the entire follow-up. None of the patients had co-existing asthma at entry and, for simplicity, we assumed that asthmatic disease could not complicate ARC already from the first year.

- SCIT: the therapy is administered for a maximum of three years, in combination with symptomatic therapy. After a maximum of three Markov cycles (i.e. $t = 0, 1, 2$), the immuno-logic treatment is stopped, and patients continue symptomatic drug therapy alone. AIT discontinuation can occur already from the first Markovian cycle ($t = 0$). Also in this case,

patients continue symptomatic therapy over the entire follow-up. As in the case of the `no AIT` strategy, asthma cannot complicate ARC already from the first year.

- `SLIT`: the assumptions for the `SCIT` strategy shall apply once the necessary changes have been made.

## Transition matrices

Transition probabilities between states can be described by a square $|\mathcal{S}_i| \times |\mathcal{S}_i|$ stochastic matrix $T$ (whose rows sum to 1).

The evolution of the `no AIT` strategy is governed by an homogeneous Markov chain (time-invariant), having the state transition diagram shown in Fig 1 (valid for the whole time horizon considered for the simulation) corresponding to the transition matrix denoted with $T_1$.

# no AIT

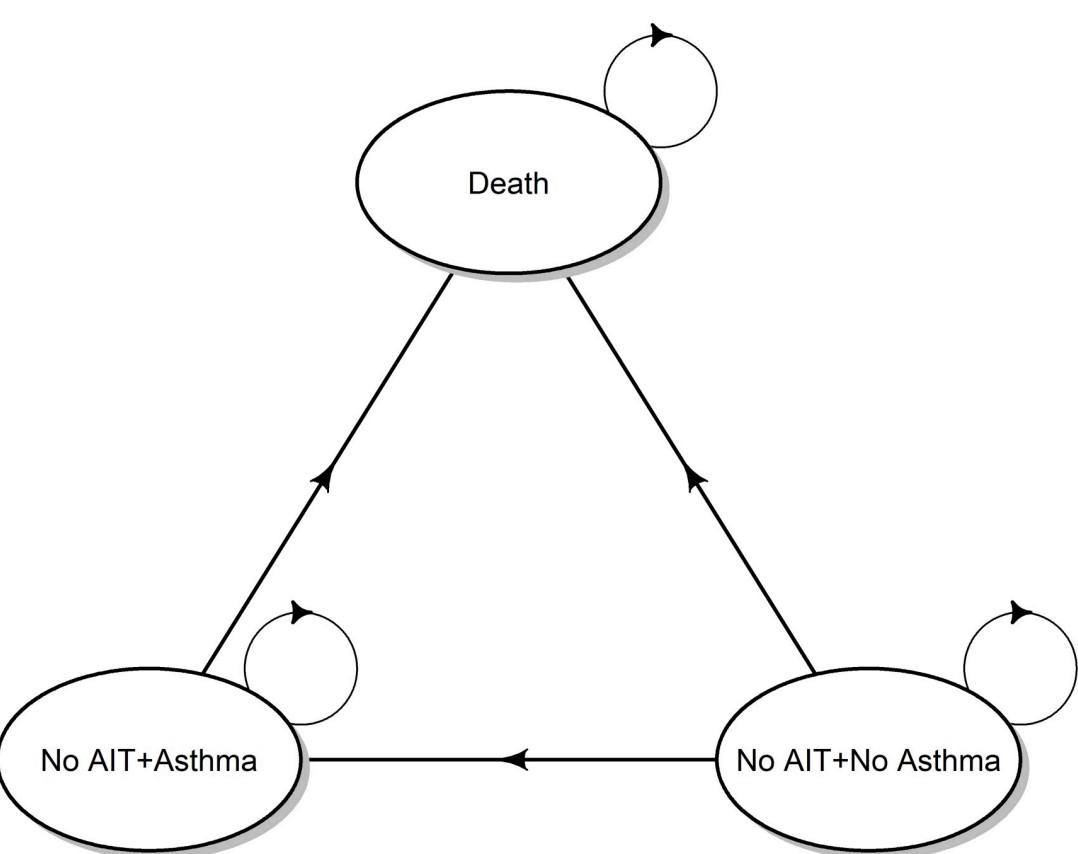

**Fig 1. Transition diagram (no AIT).** State transition diagram for the `no AIT` strategy, based on symptomatic drug therapy.

For the `no AIT` strategy, the probability of being in a given health state at the end of Markov cycle $t$ is given by:

$$\pi_{1,t} = \pi_{1,0} T_1^t \tag{1}$$

where $\pi_{1,t}$ is an $|\mathcal{S}_1|$-dimensional row vector. Consequently, the cohort is distributed among possible states according to the following vector:

$$N\pi_{1,t} \tag{2}$$

Both `SCIT` and `SLIT` strategies (or AIT for brevity's sake, when it is not relevant to distinguish between them) evolve according to a non-homogeneous Markov chain, following the same health state transitions in both cases. Transition probabilities are different between the two strategies, and the corresponding transition matrices will be denoted as $T_{2,t}$ (`SCIT`) and $T_{3,t}$ `SLIT`, respectively. The transition diagram is shown below in Fig 2.

Up to $t = 2$, among state transitions at time $t = 1$ given the state at $t = 0$ and at time $t = 2$ given the state at $t = 1$, the following changes correspond to therapy discontinuation:

- `AIT + No Asthma` → `No AIT + No Asthma`

- `AIT + Asthma` → `No AIT + Asthma`

As already highlighted before, at the end of the third cycle (year), all patients complete their immunological treatment. Therefore, any state transition at time $t = 3$ given the state at $t = 2$ must consider the fact that the probability of being in any state where the immunotherapy is administered must be set to zero. For subsequent cycles, the two states `AIT + No Asthma` and `AIT + Asthma` are no longer reachable, and thus become isolated. The underlying Markov chain becomes observationally equivalent to that defined for the `no AIT` strategy. Moreover, it is worth noting that the probability distribution of states in $t = 3$ is not equal to the initial distribution $\pi_{1,0}$ of the `no AIT` strategy, but reflects the distribution between asthmatic and non-asthmatic patients that has been reached through the state transitions that occurred in the previous cycles.

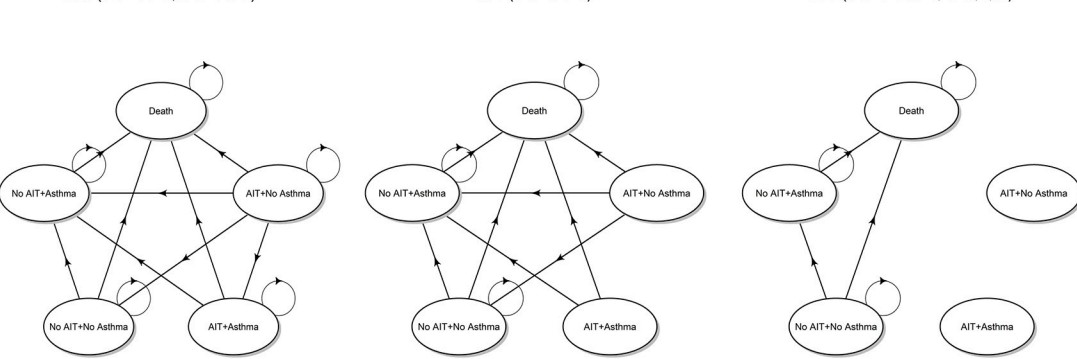

**Fig 2. Transition diagrams (AIT).** State transition diagrams for `SCIT` and `SLIT` strategies, both referred to as AIT for brevity.

For both `SCIT` and `SLIT` ($i$ = 2, 3), the probability of being in a given health state at the end of Markov cycle $t$ is given by [32]:

$$\pi_{i,t} = \pi_{i,0} \prod_{k=1}^{t} \mathrm{T}_{i,k} \qquad (3)$$

where $\mathrm{T}_{i,k}$ contains the transition probabilities for cycle $t = k$ (given the state at $t = k - 1$).

## Transition probabilities

In order to obtain transition probabilities, a literature review was performed to determine sensible model input parameters. Uncertainty originating from model specification and from uncertainty around the true value of input parameters will be discussed further in the Results and Discussion sections (see also [28] for a good introduction of the topic).

- **no AIT.** The input parameters of this strategy are not specific, and they will be used for both `SCIT` and `SLIT` as well. In particular:

- The calculation of 1-year probability of asthma onset was based on the data reported by [34], where the frequency of asthma was studied in 6461 participants, aged 20–44 years, without asthma at baseline. In particular (page 1052) it is reported that: "*The probability of having asthma at the end of follow-up (mean 8.8 years) was 4.0% in patients with allergic rhinitis*". For this sub-population of 1217 patents at risk with ARC, the average age was 33.4 years. In this way, both the length of the follow-up and the average age were consistent with our setting. If $p_a$ = 0.040 is the probability that patients without asthma at baseline have asthma at 8.8-year, the 1-year probability can be easily computed assuming that the incidence rate is constant over each time period [35]. Under this assumption, the annual incidence rate is:

$$r_{a(1yr)} = -\frac{\log(1 - p_a)}{t}$$

and, finally, the time-frame was changed obtaining the 1-year incidence probability in the following way:

$$p_{a(1yr)} = 1 - \exp(-r_{a(1yr)}) \approx 0.00463$$

- The probability of death within 1 year for all causes was obtained from the Italian National Institute of Statistics (ISTAT) mortality tables for general mortality [36]. The average age (35.9 years) of the patients included in the meta-analysis by [21] was used as reference. Considering that the studies included in the meta-analysis were published between 1999 and 2014, covering a total of 16 years, we used the probability of death within one year for all causes at age 36 in 2010 irrespective of gender, that is $p_{d(1yr)}$ = 0.00061

- The mortality among asthmatic patients was adjusted on the ground of data published by [37], who reported that mortality from all causes on a population of Finnish adults was increased among asthmatics, with an age-adjusted hazard ratio equal to $RR_{d(a)}$ = 1:49

The initial state vector, representing the distribution of the cohort at the end of the first cycle, was given by (4). As by assumption none of the patients had co-existing asthma at entry,

and asthmatic disease cannot complicate ARC already from the first year.

$$N\pi_{1,0} = \begin{array}{cc} & \begin{array}{ccc} NA_1 & A_1 & D_1 \end{array} \\ & (1000 \quad 0 \quad 0) \end{array} \tag{4}$$

Once the input parameters were suitably combined into the transition matrix, we obtained the following parametric specification. The specific values of transition probabilities were superimposed on the transition diagram shown below in Fig 3:

$$T_1 = \begin{array}{c} \\ NA_1 \\ A_1 \\ D_1 \end{array} \begin{array}{c} \begin{array}{ccc} NA_1 & A_1 & D_1 \end{array} \\ \begin{pmatrix} 1 - (\bullet) & p_{a(1yr)} & p_{d(1yr)} \\ 0 & 1 - (\bullet) & RR_{d(a)}p_{d(1yr)} \\ 0 & 0 & 1 \end{pmatrix} \end{array} \tag{5}$$

- **SCIT.** Both clinical trials and real-life settings have demonstrated that AIT course is frequently problematic due to poor treatment adherence, hence jeopardizing the immunological effects that underlie the clinical outcome [38–41]. In light of this, input parameters must keep adherence into account due to its relevant effect. On the other hand, there is an accumulating body of evidence that a well-conducted immunological treatment can reduce the development of asthma [42, 43]. Consequently:

- For calculating the probability of discontinuing SCIT, we used the data from [40], Table 3, considering the following setting: pollen-preseasonal, age $\geq$ 18 (adults), $N$ = 38576. Based on the number of patients who completed the first year of therapy but discontinued during the second year, and those who completed two years of therapy but discontinued during the third, we obtained the following discontinuation probabilities for each of the 3 years of treatment: $p_{d(1yr)} = 0$, $p_{dr(2yr_s)} = 0.53$, $p_{dr(3yr_s)} = 0.45$. According to these data, the adherence to SCIT is maximum during the first year of therapy (i.e. the probability of discontinuation is zero).

- According to [33, 44, 45], the relative risk of developing asthma (AIT vs. symptomatic treatment) was set as $RR_{a(AIT)}$ = 0.505. This input parameter is not specific to SCIT; it is valid for SLIT as well.

As for no AIT strategy, the initial state vector is given by:

$$N\pi_{2,0} = \begin{array}{cc} & \begin{array}{ccccc} ANA_2 & AA_2 & NANA_2 & NAA_2 & D_2 \end{array} \\ & (1000 \quad 0 \quad 0 \quad 0 \quad 0) \end{array} \tag{6}$$

In modeling the dynamics of $\mathcal{S}$-valued time-inhomogeneous Markov chains, we need to specify the sequence of (one-step) transition matrices. Transition probabilities at time $t = 1$

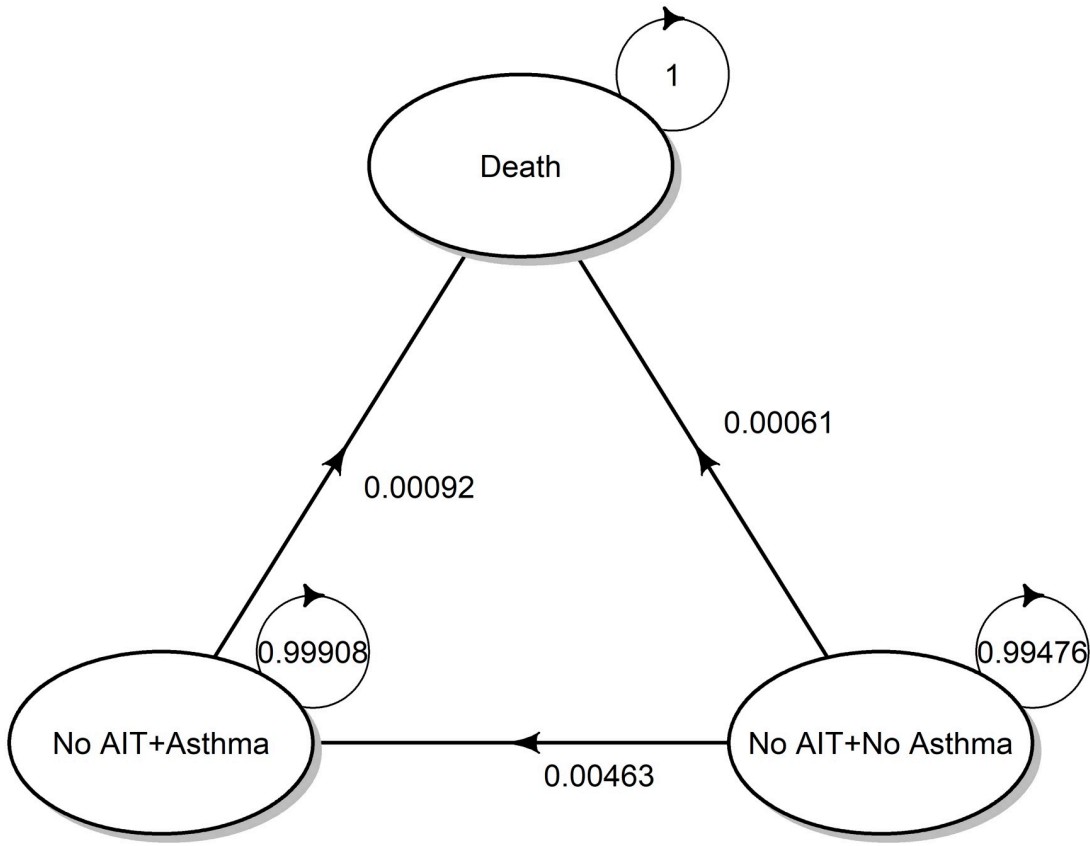

# no AIT

**Fig 3. Transition probabilities.** State transition diagram for the `no AIT` strategy with superimposed transition probabilities, calculated according to transition matrix 5.

given the state occupied at $t = 0$ are collected in the $T_{2,1}$ matrix:

$$T_{2,1} = \begin{array}{c} \\ ANA_2 \\ AA_2 \\ NANA_2 \\ NAA_2 \\ D_2 \end{array} \begin{array}{ccccc} ANA_2 & AA_2 & NANA_2 & NAA_2 & D_2 \end{array} \\ \left( \begin{array}{ccccc} 1-(\bullet) & RR_{a(AIT)}p_{a(1yr)} & p_{dr(2yr_s)} & p_{dr(2yr_s)}\left(RR_{a(AIT)}p_{a(1yr)}\right) & p_{d(1yr)} \\ 0 & 1-(\bullet) & 0 & p_{dr(2yr_s)} & RR_{d(a)}p_{d(1yr)} \\ 0 & 0 & 1-(\bullet) & p_{a(1yr)} & p_{d(1yr)} \\ 0 & 0 & 0 & 1-(\bullet) & RR_{d(a)}p_{d(1yr)} \\ 0 & 0 & 0 & 0 & 1 \end{array} \right) \quad (7)$$

The transition probability in cell $(1, 4)$ deserves further explanation. Such probability describes the case where a discontinuation of therapy and a newly onset asthmatic disease

occur together. A causal relationship between these two events is likely. But in the absence of data strongly supporting this hypothesis, we can treat the two events as conditionally independent. Therefore, we can write down the following approximation:

$$pr(\text{ANA}_2 \rightarrow \text{NAA}_2) \quad = \quad pr(\text{AIT + No Asthma} \rightarrow \text{No AIT + Asthma}) = \tag{8}$$

$$= \quad pr(\text{No AIT + Asthma} \mid \text{AIT + No Asthma}) = \tag{9}$$

$$= \quad pr(\text{No AIT} \mid \text{AIT + No Asthma}) \times \tag{10}$$

$$\times \quad pr(\text{Asthma} \mid \text{AIT + No Asthma}) = \tag{11}$$

$$= \quad p_{dr(2yr_s)} \times \left( RR_{a(AIT)} p_{a(1yr)} \right) \tag{12}$$

It should be also highlighted that we are assuming that the protective effect of the immunologic treatment on asthma is active only after one year of therapy. After three years of treatment, the protective effect becomes permanent for the whole horizon of the simulation; on the other hand, those who prematurely interrupt the immunologic therapy lose the protective effect, and the 1-year asthma probability almost doubled from $RR_{\text{a (AIT)}} \, p_{\text{a (1yr)}}$ to $p_{\text{a (1yr)}}$.

Transition probabilities at $t = 2$ given the state occupied at $t = 1$ have a similar structure, once the necessary changes have been made:

$$T_{2,2} = \begin{array}{c} \\ \text{ANA}_2 \\ \text{AA}_2 \\ \text{NANA}_2 \\ \text{NAA}_2 \\ \text{D}_2 \end{array} \begin{array}{ccccc} \text{ANA}_2 & \text{AA}_2 & \text{NANA}_2 & \text{NAA}_2 & \text{D}_2 \\ \left( 1 - (\bullet) \right. & RR_{a(AIT)}p_{a(1yr)} & p_{dr(3yr_s)} & p_{dr(3yr_s)}\left(RR_{a(AIT)}p_{a(1yr)}\right) & p_{d(1yr)} \\ 0 & 1 - (\bullet) & 0 & p_{dr(3yr_s)} & RR_{d(a)}p_{d(1yr)} \\ 0 & 0 & 1 - (\bullet) & p_{a(1yr)} & p_{d(1yr)} \\ 0 & 0 & 0 & 1 - (\bullet) & RR_{d(a)}p_{d(1yr)} \\ 0 & 0 & 0 & 0 & \left. 1 \right) \end{array} \tag{13}$$

As stated before, transition probabilities at $t = 3$ must keep into account that patients terminate the immunotherapy, and thus the probability of reaching a state where AIT is administered must be zero:

$$T_{2,3} = \begin{array}{c} \\ \text{ANA}_2 \\ \text{AA}_2 \\ \text{NANA}_2 \\ \text{NAA}_2 \\ \text{D}_2 \end{array} \begin{array}{ccccc} \text{ANA}_2 & \text{AA}_2 & \text{NANA}_2 & \text{NAA}_2 & \text{D}_2 \\ \left( 0 \right. & 0 & 1 - (\bullet) & RR_{a(AIT)}p_{a(1yr)} & p_{d(1yr)} \\ 0 & 0 & 0 & 1 - (\bullet) & RR_{d(a)}p_{d(1yr)} \\ 0 & 0 & 1 - (\bullet) & 0.00403 & p_{d(1yr)} \\ 0 & 0 & 0 & 1 - (\bullet) & RR_{d(a)}p_{d(1yr)} \\ 0 & 0 & 0 & 0 & \left. 1 \right) \end{array} \tag{14}$$

Transition probability in cell (3, 4) is $\Pr(\text{No AIT + No Asthma} \rightarrow \text{No AIT + Asthma})$, that is the 1-year probability of developing asthma. Up to $t = 2$ a patient reaches a `No AIT` state because she/he has interrupted immunotherapy. From $t = 3$ onwards, a patient who has not prematurely interrupted the treatment has completed a full immunotherapy cycle in $t = 2$, and, with probability 1, she/he moves towards any state labeled `No AIT`. For patients who have not completed a full cycle, the 1-year probability of asthma is $p_{\text{a (1yr)}}$, whereas patients who have completed their therapy are provided with long-term protection, and their 1-year risk of asthma drops to $RR_{\text{d (a)}} \, p_{\text{d (1yr)}}$. From the data reported in [40], Table 3, 26% of patients carry out SCIT for at least three years, and thus the transition probability in cell (3, 4)

is calculated as the following weighted average:

$$\Pr(\texttt{No AIT} + \texttt{No Asthma} \to \texttt{No AIT} + \texttt{Asthma}) =$$
$$= 0.74\, p_{\texttt{a(1yr)}} + 0.26 RR_{\texttt{a(AIT)}} p_{\texttt{a(1yr)}} \approx 0.00403 \tag{15}$$

For $t = 4, 5, \ldots$ transition probabilities are collected in the matrices $\mathrm{T}_{2,t}$. States `AIT + no Asthma` and `AIT + Asthma` become isolated, and patients of the cohort cannot reach them anymore.

$$
\mathrm{T}_{2,t} = 
\begin{array}{c}
\\
\texttt{ANA}_2 \\
\texttt{AA}_2 \\
\texttt{NANA}_2 \\
\texttt{NAA}_2 \\
\texttt{D}_2
\end{array}
\overset{\begin{array}{ccccc} \texttt{ANA}_2 & \texttt{AA}_2 & \texttt{NANA}_2 & \texttt{NAA}_2 & \texttt{D}_2 \end{array}}{
\left(
\begin{array}{ccccc}
1 & 0 & 0 & 0 & 0 \\
0 & 1 & 0 & 0 & 0 \\
0 & 0 & 1-(\bullet) & 0.00403 & p_{\texttt{d(1yr)}} \\
0 & 0 & 0 & 1-(\bullet) & RR_{\texttt{d(a)}} p_{\texttt{d(1yr)}} \\
0 & 0 & 0 & 0 & 1
\end{array}
\right)}
\tag{16}
$$

Transition diagrams with superimposed transition probabilities are shown in Fig 4.

• **SLIT.** There are a few differences with the preceding strategy that are worth examining in detail:

- From [40], using Table 3 with parameters: pollen, age $\geq$ 18 (adults), $N = 570$, we obtained the following discontinuation probabilities: $p_{\texttt{dr(1yr)}} = 0.52$, $p_{\texttt{dr(2yr}_s)} = 0.49$, $p_{\texttt{dr(3yr}_s)} = 0.39$. There is an apparently lower adherence to therapy than SCIT, that has been reported in other studies (see [46] for recent data).

The initial state vector in $t = 0$ for the `SLIT` arm keeps into account that only 52% stay in the `AIT + No Asthma` state, while the remaining 48% are in `No AIT + No Asthma` state, as these patients started the therapy but discontinued it already in the first year. This has a deep influence on cost calculation already from the first Markovian cycle. Hence the initial state vector is:

$$
N\pi_{3,0} \quad
\overset{\begin{array}{ccccc} \texttt{ANA}_3 & & \texttt{AA}_3 & \texttt{NANA}_3 & \texttt{NAA}_3 & \texttt{D}_3 \end{array}}{
\left(1000 - (1-p_{\texttt{dr(1yr)}}) \quad\quad 0 \quad\quad 1000 p_{\texttt{dr(1yr)}} \quad\quad 0 \quad\quad 0\right)}
\tag{17}
$$

Transition matrices for $t = 1$ and $t = 2$ are formally identical to (7) and (13), that is $\mathrm{T}_{3,1} \equiv \mathrm{T}_{2,1}$, $\mathrm{T}_{3,2} \equiv \mathrm{T}_{2,2}$, after modifying what was needed to be changed. For $t = 3$ and $t = 4, 5, \ldots, \ldots$ we have:

$$
\mathrm{T}_{3,3} = 
\begin{array}{c}
\\
\texttt{ANA}_3 \\
\texttt{AA}_3 \\
\texttt{NANA}_3 \\
\texttt{NAA}_3 \\
\texttt{D}_3
\end{array}
\overset{\begin{array}{ccccc} \texttt{ANA}_3 & \texttt{AA}_3 & \texttt{NANA}_3 & \texttt{NAA}_3 & \texttt{D}_3 \end{array}}{
\left(
\begin{array}{ccccc}
0 & 0 & 1-(\bullet) & RR_{\texttt{a(AIT)}} p_{\texttt{a(1yr)}} & p_{\texttt{d(1yr)}} \\
0 & 0 & 0 & 1-(\bullet) & RR_{\texttt{d(a)}} p_{\texttt{d(1yr)}} \\
0 & 0 & 1-(\bullet) & 0.00428 & p_{\texttt{d(1yr)}} \\
0 & 0 & 0 & 1-(\bullet) & RR_{\texttt{d(a)}} p_{\texttt{d(1yr)}} \\
0 & 0 & 0 & 0 & 1
\end{array}
\right)}
\tag{18}
$$

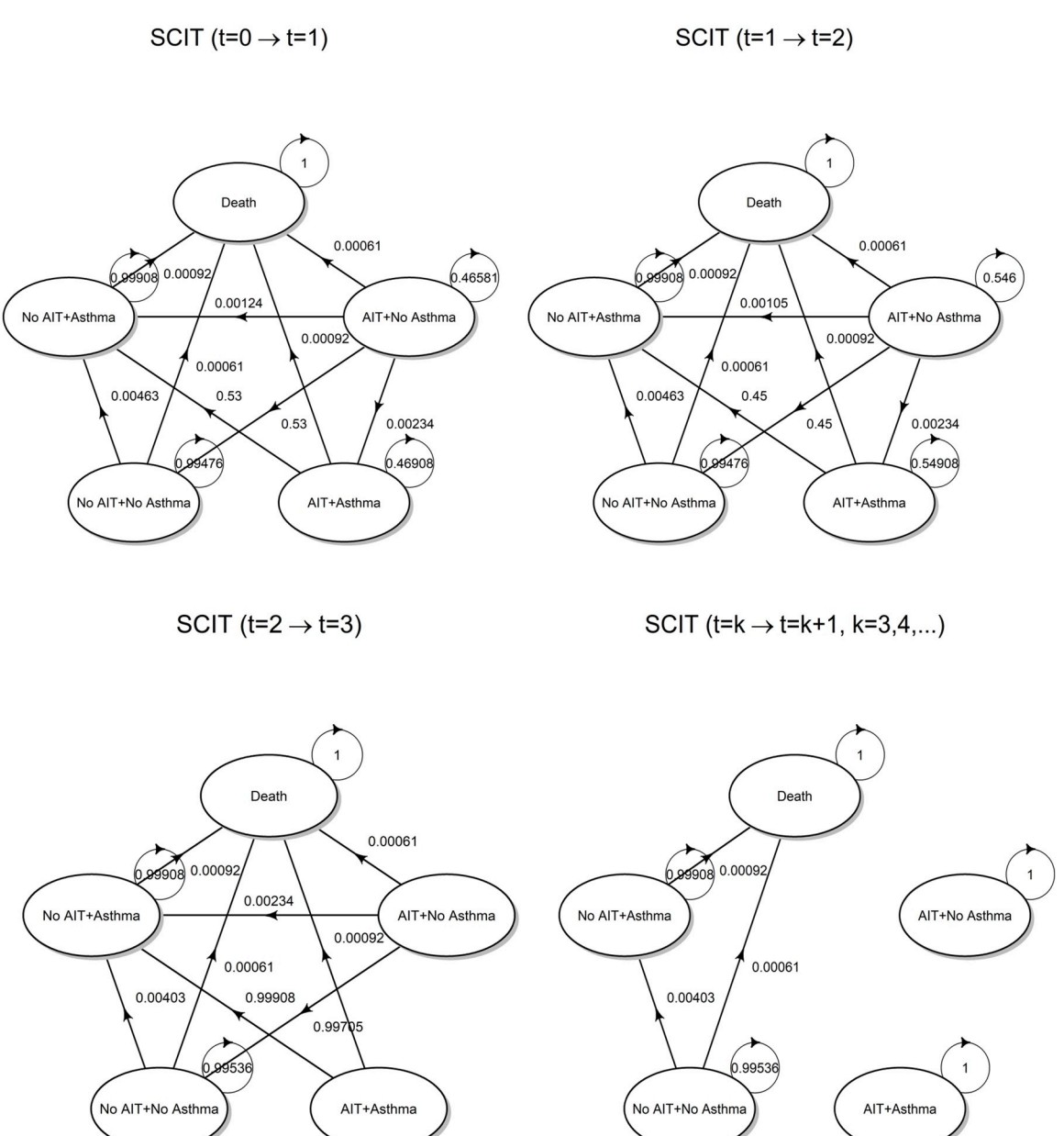

**Fig 4. Transition probabilities.** State transition diagram for the `SCIT` strategy with superimposed transition probabilities, calculated according to transition matrices (7), (13), (14) and (16).

$$
T_{3,k} = 
\begin{array}{c}
\\
\text{ANA}_3 \\
\text{AA}_3 \\
\text{NANA}_3 \\
\text{NAA}_3 \\
\text{D}_3
\end{array}
\begin{array}{ccccc}
\text{ANA}_3 & \text{AA}_3 & \text{NANA}_3 & \text{NAA}_3 & \text{D}_3 \\
\left(\begin{array}{ccccc}
1 & 0 & 0 & 0 & 0 \\
0 & 1 & 0 & 0 & 0 \\
0 & 0 & 1-(\bullet) & 0.00428 & p_{\text{d(1yr)}} \\
0 & 0 & 0 & 1-(\bullet) & RR_{\text{d(a)}} p_{\text{d(1yr)}} \\
0 & 0 & 0 & 0 & 1
\end{array}\right)
\end{array}
\tag{19}
$$

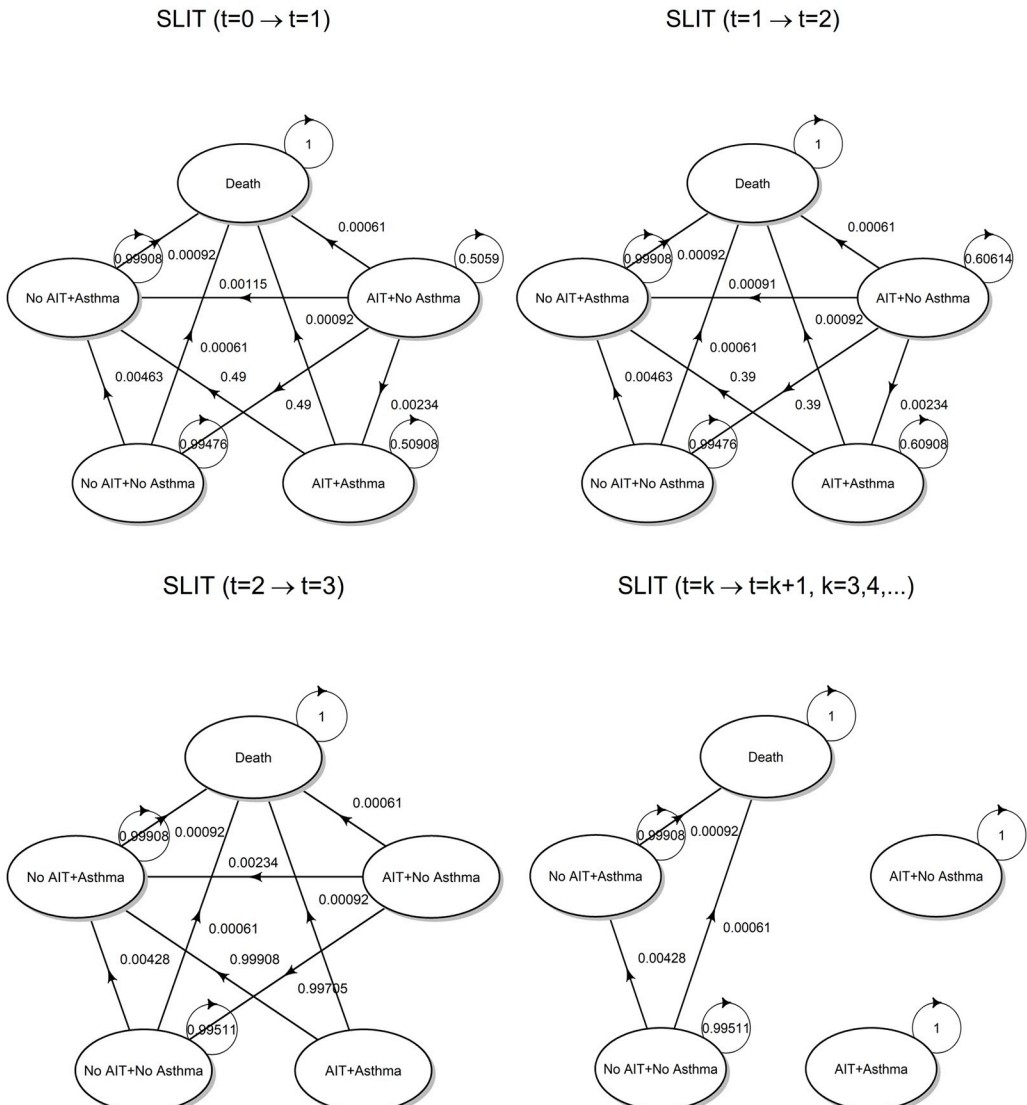

**Fig 5. Transition probabilities.** State transition diagram for the `SLIT` strategy with superimposed transition probabilities, calculated according to transition matrices defined above.

As [40] reports that 15% of patients completed at least three years of therapy, the transition probability in cell (3, 4) can be calculated as:

$$\mathrm{Pr}(\texttt{No\ AIT} + \texttt{No\ Asthma} \rightarrow \texttt{No\ AIT} + \texttt{Asthma}) =$$
$$= 0.85\, p_{\mathrm{a(1yr)}} + 0.15 RR_{\mathrm{a(AIT)}} p_{\mathrm{a(1yr)}} \approx 0.00428 \tag{20}$$

Finally, the specific values assumed by transition probabilities are shown in Fig 5.

## Utilities

Utilities for health states are based on preferences for the different health states, in the sense that the more desirable (i.e. less severe) health states will receive greater weight. Utilities are measured on a cardinal scale 0–1, where 0 indicates death and 1 full health [47].

**Table 1. For each integer RTSS, columns 2 to 5 show the probability distribution on symptom severity, column 6 ($u_i$) reports the RSUI calculated according to 21. Finally, column 7 ($u_{i,\text{interp}}$) shows the model-based RSUI (see the text for a full description).**

| RTSS | None | Mild | Moderate | Severe | $u_i$ | $u_{i,\text{interp}}$ |
|------|------|------|----------|--------|-------|------------------------|
| 0 | 1.0000 | 0.0000 | 0.0000 | 0.0000 | 1.0000 | 0.9999 |
| 1 | 0.8333 | 0.1667 | 0.0000 | 0.0000 | 0.9440 | 0.9444 |
| 2 | 0.7143 | 0.2381 | 0.0476 | 0.0000 | 0.8873 | 0.8864 |
| 3 | 0.6250 | 0.2679 | 0.0893 | 0.0179 | 0.8255 | 0.8255 |
| 4 | 0.5417 | 0.2917 | 0.1250 | 0.0417 | 0.7640 | 0.7638 |
| 5 | 0.4676 | 0.3009 | 0.1620 | 0.0694 | 0.7027 | 0.7029 |
| 6 | 0.4018 | 0.3006 | 0.1935 | 0.1042 | 0.6442 | 0.6437 |
| 7 | 0.3399 | 0.2961 | 0.2215 | 0.1425 | 0.5864 | 0.5864 |
| 8 | 0.2839 | 0.2839 | 0.2473 | 0.1850 | 0.5314 | 0.5310 |
| 9 | 0.2328 | 0.2672 | 0.2672 | 0.2328 | 0.4773 | 0.4775 |
| 10 | 0.1850 | 0.2473 | 0.2839 | 0.2839 | 0.4265 | 0.4261 |
| 11 | 0.1425 | 0.2215 | 0.2961 | 0.3399 | 0.3769 | 0.3769 |
| 12 | 0.1042 | 0.1935 | 0.3006 | 0.4018 | 0.3301 | 0.3298 |
| 13 | 0.0694 | 0.1620 | 0.3009 | 0.4676 | 0.2846 | 0.2848 |
| 14 | 0.0417 | 0.1250 | 0.2917 | 0.5417 | 0.2421 | 0.2419 |
| 15 | 0.0179 | 0.0893 | 0.2679 | 0.6250 | 0.2006 | 0.2007 |
| 16 | 0.0000 | 0.0476 | 0.2381 | 0.7143 | 0.1612 | 0.1608 |
| 17 | 0.0000 | 0.0000 | 0.1667 | 0.8333 | 0.1210 | 0.1212 |
| 18 | 0.0000 | 0.0000 | 0.0000 | 1.0000 | 0.0792 | 0.0790 |

A quantitative index of ARC severity is the Rhinoconjunctivitis Total Symptom Score (RTSS; [48]), based on patient reported outcomes. RTSS considers the six symptoms most commonly associated with pollinosis (sneezing, rhinorrhea, nasal pruritus, nasal congestion, ocular pruritus and watery eyes). The self-attributed score to each symptom varies from 0 to 3, namely: 0 = none, 1 = mild, 2 = moderate, 3 = severe. The total of the scores attributed to these six symptoms results in a daily RTSS that can vary from 0 to 18. The final RTSS is calculated as the mean of the total daily scores recorded throughout the whole pollen season.

The utilities associated with seasonal ARC have been estimated in [49], who developed a multi-attribute utility index for symptoms of this disease, called RSUI (Rhinitis Symptom Utility Index). This index has been constructed based on the interplay between the two existing standard approaches to assign an utility score to an health state, i.e. the VAS (Visual Analog Scale) and SG (Standard Gamble) methods [50, 51]. Mapping RTSS one-to-one to RSUI was largely based on the same procedure used in [33], as communicated to our study group by one of the authors (K.Y. Westerhout, personal communication, 2016-03-10. Full details of the mapping algorithm used are available upon request).

In synthesis, for each possible value of the RTSS score, we were able to calculate the probability distribution of symptom severity, where severity was categorized according to four levels: 0 = none, 1 = mild, 2 = moderate and 3 = severe, as shown in Table 1. For example, in a randomly chosen patient having RTSS = 1, there is an 83% probability that any of the symptoms is present with a degree of severity equal to 0 (or, in other words, the symptom is not present), and a 17% probability that the same symptom has a degree of severity equal to 1 (mild). Finally, each RTSS integer score was mapped in a one-to-one fashion to an utility score using the multi-attribute multiplicative function defined in [49]:

$$u_i = 1.198(\bar{s}_1 \times \bar{s}_2 \times \bar{s}_3 \times \bar{s}_4 \times \bar{s}_5 \times \bar{s}_6) - 0.198 \qquad (21)$$

**Table 2. Age- and age- and asthma-adjusted utilities for grass-pollen induced season ARC, for each of the three therapeutic regimes.**

| Age classes | 25–34 | 35–49 | 50–64 | >65 |
|---|---|---|---|---|
| *Age-adjusted utilities* | | | | |
| no AIT FE | 0.7558 | 0.7402 | 0.7013 | 0.7091 |
| no AIT RE | 0.7516 | 0.7361 | 0.6973 | 0.7051 |
| SCIT FE | 0.8254 | 0.8083 | 0.7658 | 0.7743 |
| SCIT RE | 0.9228 | 0.9037 | 0.8562 | 0.8657 |
| SLIT FE | 0.8057 | 0.7891 | 0.7475 | 0.7578 |
| SLIT RE | 0.8014 | 0.7849 | 0.7436 | 0.7518 |
| *Age- and asthma-adjusted utilities* | | | | |
| SYMPT. FE | 0.5576 | 0.5461 | 0.5174 | 0.5232 |
| SYMPT. RE | 0.5545 | 0.5431 | 0.5145 | 0.5202 |
| SCIT FE | 0.6090 | 0.5964 | 0.5650 | 0.5713 |
| SCIT RE | 0.6808 | 0.6668 | 0.6317 | 0.6387 |
| SLIT FE | 0.5945 | 0.5822 | 0.5515 | 0.5576 |
| SLIT RE | 0.5913 | 0.5791 | 0.5486 | 0.5547 |

for $i$ = 0, 1, 2, . . ., 18. For a given RTSS, coefficients $\bar{s}_1, \bar{s}_2, \bar{s}_3, \bar{s}_4, \bar{s}_5$, and $\bar{s}_6$, were obtained by taking, for each of the six symptoms used in the calculation of the RTSS, a weighted average of utility scores in Table 2 by [49] (averaged by symptom severity), weighted by the corresponding probability distribution over symptom severity reported in Table 1. Utilities provided by [49] considered only five health states (symptoms), whereas the RTSS is based on six symptoms. For this reason, in order to create a correspondence between the two systems, the column present in the above-mentioned Table 2, named "itchy eyes", was duplicated and named "watery eyes".

Utility values $u_i$ can range from a maximum value of 1, when no symptoms are present, to a minimum value of about 0.07 in the worst possible case. However, we introduced some minor variations considering that the RTSS score used in pharmacoeconomic studies it is hardly ever an integer number. On the contrary, it is often an average value estimated from meta-analyses over selected populations. Therefore, the first operation that is usually carried out is rounding such average score to the nearest integer. As estimated by [52], the Minimally Important Difference (MID, i.e. the smallest improvement considered worthwhile by a patient) corresponds to a reduction of about 1.1–1.3 points of the RTSS in patients with grass pollen induced ARC, a threshold that can be conceivably rounded to 1 in according to [52]. If we consider, for example, an average RTSS score of 4.51 in the treatment group, rounded to RTSS = 5, and an average RTSS score 4.49 in the control group, rounded to RTTS = 4, we introduce a fictitious clinical improvement in the treatment group (according to the MID estimate proposed by [52]) which is not actually present in the data. To prevent such rounding bias, we considered an interpolation procedure capable of mapping RTSS scores to RSUI also in case of a non-integer RTSS score. The procedure consists of the following steps:

- For each degree of symptom severity (SSev: 0 = none, 1 = mild, 2 = moderate, 3 = severe), we overfitted a statistical relationship between $X$ = RTSS and $Y = p_{SSev}$, where $p_{SSev}$ is the corresponding probability in the Table 1, using the following polynomial regression model:

$$p_{SSev} = \beta_0 + \beta_1 X + \cdots + \beta_p X^p + \text{noise}$$

By setting $p$ = 9, we were able to make the overall root mean square prediction error less

than $10^{-3}$. In this way, theoretical probabilities estimated by the polynomial regression model in correspondence of an integer RTSS score are almost indistinguishable from the respective empirical probabilities reported in 1. Moreover, the polynomial model can be used to interpolate probabilities in correspondence of any sensible (possibly non-integer) RTSS as follows:

$$\hat{p}_{\text{SSev}} = \hat{\beta}_0 + \hat{\beta}_1 \text{RTSS} + \cdots + \hat{\beta}_p \text{RTSS}^p$$

- As the four the probabilities in each row, one for each severity level, must sum to 1, the model-based probabilities for 0 = none, 1 = mild and 2 = moderate can be obtained by direct interpolation, while those relative to 3 = severe can be calculated by difference:

$$\hat{p}_{\text{severe}} = 1 - \hat{p}_{\text{none}} - \hat{p}_{\text{mild}} - \hat{p}_{\text{moderate}}$$

- Finally, on the basis of those interpolated probabilities, we can estimate the RSUI score in correspondence of any possibly non-integer RTSS. For example, for RTSS = 4.50 we get $u_{\text{interp}}$ = 0.7332, which is roughly equal to the midpoint of the interval (0.7027, 0.7640). For reader's convenience, Table 1 also reports the model-based estimated utilities in correspondence of integer RTSS scores.

Having derived a reasonable algorithm for converting a condition-specific symptom score into utilities, we used the following data sources for attaching utilities to health states:

- SCIT, the experimental arm of the meta-analysis 3 (Fig 1) in [22] on the efficacy of SCIT versus placebo for grass-pollen induced seasonal ARC, based on standardized mean differences (SMD) for symptom score (SS).

- SLIT, the experimental arm of the meta-analysis B (Fig 1) in [21] on the efficacy of SLIT versus placebo for grass-pollen induced seasonal ARC, based on mean differences (MD) for SS.

- no AIT, the control arm of the latter meta-analysis.

From [21], in order to maintain consistency with the estimated pooled effect size, the average symptom scores in each of the two arms were calculated as:

$$\text{average SS in the experimental arm} = \bar{x}^{(e)} = \frac{\sum_{i=1}^{K} w_i^{(e)} \bar{x}_i^{(e)}}{\sum_{i=1}^{K} w_i^{(e)}} \quad (22)$$

$$\text{average SS in the control arm} = \bar{x}^{(c)} = \frac{\sum_{i=1}^{K} w_i^{(c)} \bar{x}_i^{(c)}}{\sum_{i=1}^{K} w_i^{(c)}} \quad (23)$$

where:

- $\bar{x}_i^{(e)}$ is the mean of the $i$th study included in the experimental arm.

- $\bar{x}_i^{(c)}$ is the mean of the $i$th study included in the control arm.

- $w_i^{(\cdot)}$ are the respective weights, which differ according to whether either a fixed or a random effect model (FE and RE, respectively) is used for combining data from multiple study [53].

The obtained values were, respectively:

$$
\begin{aligned}
\bar{x}^{(c)} &= \begin{cases} 3.7507, & \text{FE model} \\ 3.8208, & \text{RE model} \end{cases} \quad \rightarrow \quad \text{average No AITRTSS} \\
\bar{x}^{(e)} &= \begin{cases} 2.9176, & \text{FE model} \\ 2.9891, & \text{RE model} \end{cases} \quad \rightarrow \quad \text{average SLIT RTSS}
\end{aligned}
\tag{24}
$$

All the studies included in [21] used the RTSS ranging from 0 to 18 as the outcome measure, and hence, the obtained average scores were amenable to transformation into health states utilities.

Unfortunately, the same procedure could not be applied to the meta-analysis in [22], because effect sizes are expressed on a standardized scale. A sensible solution consists in back-transforming the standardized pooled effect size to a non-standardized scale, thus comparable with that used in the previous meta-analysis. The basic idea, expressed in [54], section 12.6.4, is to use a reference standard deviation for the particular measurement scale on which we want to back-transform the standardized pooled effect size as follows:

$$
\hat{y}^{(e)} = \bar{x}^{\text{typical}} + (SMD \times SD^{\text{typical}})
\tag{25}
$$

where:

- $\hat{y}^{(e)}$ is the unstandardized average score for the experimental arm, back-transformed to the scale of measurement of interest (in our case represented by the RTSS).

- $\bar{x}^{\text{typical}}$ represents the 'typical' baseline average level of the measurement scale on which we want to back-transform. Therefore, the most reasonable choice is:

$$
\bar{x}^{\text{typical}} = \bar{x}^{(c)} = \begin{cases} 3.7507, & \text{FE model} \\ 3.8208, & \text{RE model} \end{cases}
\tag{26}
$$

- $SD^{\text{typical}}$ represents the 'typical' standard deviation of the measurement scale on which we want to back-transform. In [54] it is suggested that "... *the standard deviation could be obtained as the pooled standard deviation of baseline scores in one of the studies*.". As it is not apparent which study should be used for such purpose, we calculated $SD^{\text{typical}}$ in the following way:

$$
SD^{\text{typical}} = \frac{\sum_{i=1}^{K} w_i^{(c)} s_i^{(c)}}{\sum_{i=1}^{K} w_i^{(c)}}
$$

where $s_i^{(c)}$ is the sample standard deviation for the $i$th study in the control arm of the previous meta-analysis. In other words, rather than calculating a pooled average, weighted by degrees of freedom as usual (which, however, requires that the variances at population level are the same for all the studies), we have calculated a weighted average using the weights from the estimated model. This procedure allowed to reduce the weight of those studies that had a higher sample variability (which was likely due to a high heterogeneity of the included

patients). With this choice, we obtained the following values:

$$SD^{\text{typical}} = \begin{cases} 3.1012, & \text{FE model} \\ 3.1871, & \text{RE model} \end{cases}$$

- *SMD* is the standardized pooled effect size (pooled) of the SLIT meta-analysis, equal to -0.38 for the FE model and -0.92 for the RE model, respectively.

Using Eq (25) we obtained the following unstandardized scores:

$$\hat{y}^{(e)} = \begin{cases} 2.5876, & \text{FE model} \\ 0.8768, & \text{RE model} \end{cases} \quad \rightarrow \quad \text{average SCIT RTSS} \tag{27}$$

After transforming the average RTSS into utilities, we obtained the following baseline values: no AIT FE = 0.7792; no AIT RE = 0.7748; SCIT FE = 0.8509; SCIT RE = 0.9513; SLIT FE = 0.8306; SLIT RE: 0.8262.

These utilities refer to a single condition. When deemed appropriate, heterogeneity between individuals belonging to different age groups can be introduced into the model using a multi-utility multiplicative function as in [55], that is:

$$u_{AB} = u_A \times u_B \tag{28}$$

where $u_A$ is the utility of a single health condition and $u_B$ indicates the average utility of age class B in the absence of any pathological dimensions. In our case, $u_B$ values were obtained from the utility scores valid for Spain as reported in [56]. In a similar way, we also adjusted for the co-existence of asthma, using the utility of 0.7378 attached by [57] to patients in whom "allergic rhinitis and mild allergic asthma", characterized by intermittent use of beta-agonists, was present. We finally obtained the following sets of utilities (Table 2), that have a perfect one-to-one correspondence with the health states included in our Markov model. By way of example (using the FE-based utilities): $u(\text{NA}_1) = 0.7402$, $u(\text{NANA}_2) = 0.8083$, $u(\text{AA}_3) = 0.5822$.

## Costs

The cost of the medical resources used in the simulation was obtained, directly or indirectly, from various data sources. None of the patients included in the simulated cohort was hospitalized. We used a societal perspective [58], by quantifying direct costs due to drug or medical resource consumption, regardless whether these costs were paid by patients or by the provider of the service, that is the Italian Regions and then, ultimately, the Italian Government, as well as direct non-medical costs such as hospital transport cost, which is entirely borne by patients. We also considered indirect non-medical costs, in particular the loss of income due to the number of working hours lost as a result of immunologic treatment administration (SCIT). We did not consider intangible costs attributable the physical or psychological distress associated with the disease, given their difficult quantification [59]. All costs were given in 2018 values, except costs for asthma medications, which were given in 2010 values (although the low average annual inflation rate over the last 10 years makes this issue more apparent than real). The various cost components have been calculated as follows:

- Seasonal costs for symptomatic drug treatment, entirely borne by patients. They are obtained by summing the average seasonal cost for both Desloratadine and Budesonide:

1. Desloratidine: 5 mg orally once a day per os, the daily cost is 0.2055 €per 5 mg tablet (Desloratadina DOC—generic drug—4.11 €per pack, 20 tablets). To estimate the average daily consumption of Desloratidine during the entire pollen season we used the data from [60, 61]; in particular, [61] reported an average consumption 17.9 tablets per season in the control group and 12.2 tablets per season in the treatment group (treated with SLIT for ARC), while [60] reported that the above mentioned difference between the two groups corresponded to a reduction in the medication score from 2.4 (control group) to 1.5 (treated group, undergone SLIT immunotherapy). To transform this reference average consumption into monetary values, we used a simple proportion assuming a linear relationship between the average RTSS reduction and the average Desloratadine consumption reduction during the entire pollen season. For example, using the FE model, the reduction in the RTSS between the control and the experimental group treated with SCIT was equal to $2.9176 - 3.7507 = -0.8331$. Hence the average reduction $x$ in the Desloratadine consumption for the treated group was calculated as:

$$(12.2 - 17.9) : (1.5 - 2.4) = x : (-0.8331)$$
$$(-5.7) : (-0.9) = x : (-0.8331)$$
$$\rightarrow -5.28 \text{ tablets for the entire pollen season}$$

which corresponds to a total of $17.9 - 5.28 = 12.62$ tablets of Desloratidine on average during the entire pollen season for the SCIT strategy. By repeating the same calculations for all possible combinations (SLIT and SLIT, FE and ME model), and dividing by the mean duration of a pollen season (58 days, see [60]), we obtained the average daily consumption of Desloratadine, in tablets per day, valid for the entire pollen season. For the no AIT strategy we considered the average seasonal consumption for the control arm reported by [61], that is 17.9 tablets per season, corresponding 0.31 tablets per day. The average seasonal cost was obtained by multiplying the average daily consumption for the desired season length (e.g. 60, 90 or 120 days), and calculating the minimum number of packs necessary to ensure the average seasonal consumption. By way of example, with the SCIT strategy and a season length of 60 days, the average seasonal consumption was of 10.896 tablets, and thus only 1 pack of Desloratadina DOC was charged, for an average seasonal cost of €4.11.

2. Budesonide: two 50 mcg puffs per nostril twice a day (total 200 mcg per day), the cost per inhalation is 0.0775 €with a daily cost of $0.0775 * 4 = €0.31$, 200 mcg per die (Kesol® Spray 50mcg 10ml/200D, €15.50 per 200 inhalations). We used the same procedure described above for calculating the average seasonal cost.

- Costs for medical resource use. These costs vary as simulation run unfolds:

1. no AIT

   a. 1st year ($t = 0$):

   - first access visit: €30.66 for first specialist (physician) visit copayment + €10 for additional fixed copayment to access public health services;

   - €22.91 for further specialist visit;

   - Allergy testing: €11.92 up to 7 allergens for a total of 16 allergens, corresponding to a total cost of €11.92 x 3 + €10 (additional fixed fee) = €44.86.

   b. from 2nd year onwards: no additional use of medical resources.

2. SCIT

 a. 1$^{st}$ year ($t = 0$):

- first access visit: €30.66 for first specialist visit copayment + €10 for additional fixed copayment to access public health services;

- 12 immunotherapy administrations, 4 administrations during the build-up phase (1st month) + 8 additional administrations in the course of the year. The total cost of this therapeutic schedule must be calculated in blocks of 8 administrations, at a unit cost of €11.96 for each injection, for a total of €11.96 x 8 = €92.69 + €10 (fixed fee) = €102.69. As 1 block is not enough during the first year (we have 12 administrations), we assumed that two blocks are consumed, giving a yearly total cost of 102.69 + 102.69 = €205.38.

 b. 2$^{nd}$ year ($t = 1$):

- €22.91 for further specialist visit;

- 10 immunotherapy administrations during the year. Four injections purchased the previous year are still available, thus only one new block will be purchased by patients, incurring a cost of €92.69 + €10 (fixed copayment) = €102.69.

 c. 3$^{rd}$ year ($t = 2$):

- €22.91 for further specialist visit;

- €44.86 for allergy testing;

- 10 immunotherapy administration, €102.69 as above.

 d. from 4$^{th}$ years onwards: no additional use medical resources.

3. SLIT: the structure of costs for SLIT are identical to those charged for SLIT, except that patients do not incur in cost for administering the immunotherapy (which is self-administered).
All the calculations in this section were carried out using the (Italian) health copayment amounts valid for the Apulia Region. Assuming an average duration of 20 minutes for each specialist visit, we can conclude, with a good level of approximation, that the amount of copayment for specialist visits incurred by the patients covers the remuneration that the Apulia Region must pay to the specialist physician. Therefore, in line with the societal perspective used, we do not have to charge any other cost for the specialist visits. The copayment amount incurred by the patient for allergy tests covers the entire cost of the service, and therefore also in this case we do not have to add any other costs to be borne by the National Health System. Medical cost over a follow-up of $T = 9$ years are then hereby detailed:

- Years 0-2

  - no AIT: Average = €32.81

  - SCIT: Average = €199.95

  - SLIT: Average = €63.03

- Year 3 and afterwards

  - None of the strategies incurred in any additional cost for medical resources

- Costs of immunotherapy. These costs are entirely borne by patients:

- `SCIT`: The pharmaceutical formulations that we have considered in order to determine the average annual cost were those marketed by the companies listed below. We report the average price per pack, containing 1 vial for 5 administrations; the starter pack containing the vials used for induction response has the same price:

  1. Allergofarma (Allergovit®): €265
  2. Hal-Allergy (Purethal®): €225
  3. Lofarma (Lais In®): €214.50
  4. Allergy Therapeutics (Tyrosin TU®): €199
  5. Alk-Abellò (Pangramin Ultra®): €250
     As 2 packs are needed to administer 10 injections, the average annual for SCIT is equal to: (2 * 265 + 2 * 255 + 2 * 214.50 + 2 * 199 + 2 * 250) / 5 = €473.4

- `SLIT`: The two schedules considered for cost calculation are:

  1. GRAZAX®: a continuous cycle over three years, 1 sublingual tablet per day. Retail prices of Grazax packs are, respectively:

     - GRAZAX Os 100 Liof 75 000 Sq-t: €330.08

     - GRAZAX Os 30 Liof 75 000 Sq-t: €99.02
       Thus, one year of therapy with GRAZAZ (360 tablets) costs 3×330.08 + 2×99.02 = €1,289.46

  2. ORALAIR®: 7 months per year, from 4 months before the beginning of the pollen season, 1 sublingual tablet per day. Retail prices of Oralair are:

     - ORALAIR 30 Cpr Subl 300 lr: €99.28

     - ORALAIR 90 Cpr Subl 300 lr: €297.83
       Thus, seven months of therapy with ORALAIR (210 tablets) costs 2×297.83 + 99.28 = €694.94. The cost used in the Markov simulations was the average between these two costs, which translates to an average annual cost of (1,289.46 + 694.94) / 2 = €992.2.

- Costs of asthma: costs for asthma were derived from Table 2 in [62]. Costs have been estimated on the basis of: "... *462 subjects included in the analysis, that had mild (22.3%), moderate (34.0%) or severe (43.7%) persistent asthma* ...". The average annual direct healthcare cost (given in 2010 Euros) includes specialist medical examinations, clinical and laboratory examinations, medicines and hospital admission and discharge costs, and has been estimated at €594. The average annual indirect non-healthcare cost reflects the loss of productivity due to both lost working days and limitations in non-work-related activities, estimated at €989.

- Other costs: they must be charged for `SCIT` strategy only, as its administration involves a temporary interruption of work as well as additional transport costs. The first eventuality generates indirect non-healthcare costs (of the same kind of asthma-related indirect costs), while the second generates direct non-healthcare costs that are borne entirely by patients. Using the human-capital method, indirect costs are evaluated as the loss of income that could have been produced, but which was not actually produced because of the onset of the morbid event [63]. To obtain an hourly monetary valuation of the loss of productivity, we

considered the amount of gross domestic product per capita given in 2014 values equal to €26,697.6 [64], first divided over 52 weeks and then over 35 hours per working week, giving an average hourly value of €14.66. For transport costs we assumed a public/private mix, with a cost of €5 per each injection. Indirect costs of lost productivity and costs of transport for the `SCIT` strategy are shown below (for a follow-up of $T$ = 9 years):

- Productivity

  - `Years 0-2`: Average = €312.74

  - `Year 3 and afterwards`: This strategy did not incur any additional cost

- Transport

  - `Years 0-2`: Average = 53.33

  - `Year 3 and afterwards`: This strategy did not incur any additional cost

## Results

The evolution of cohorts based on the Markovian model that we have described was simulated in `R` 3.6.1, using the `markovchain` library, v. 0.8.0 [65, 66]. For the base case we used the fixed effects model, calculating the quality adjusted life years (QALYs) as follows:

$$
\text{QALY}_t = \begin{cases} \dfrac{(12 - k_s)}{12} \times 1 + \dfrac{k_s}{12} u(\text{NA}_1) & \text{no AIT, No Asthma} \\[2ex] \dfrac{(12 - k_s)}{12} \times 1 + \dfrac{k_s}{12} u(\text{A}_1) & \text{no AIT, Asthma} \end{cases} \tag{29}
$$

$$
\text{QALY}_t = \begin{cases} \dfrac{(12 - k_s)}{12} \times 1 + \dfrac{k_p}{12} u(\text{ANA}_2) + \dfrac{k_s - k_p}{12} u(\text{NA}_1) & \text{SCIT, No Asthma} \\[2ex] \dfrac{(12 - k_s)}{12} \times 1 + \dfrac{k_p}{12} u(\text{AA}_2) + \dfrac{k_s - k_p}{12} u(\text{A}_1) & \text{SCIT, Asthma} \end{cases} \tag{30}
$$

$$
\text{QALY}_t = \begin{cases} \dfrac{(12 - k_s)}{12} \times 1 + \dfrac{k_p}{12} u(\text{ANA}_3) + \dfrac{k_s - k_p}{12} u(\text{NA}_1) & \text{SLIT, No Asthma} \\[2ex] \dfrac{(12 - k_s)}{12} \times 1 + \dfrac{k_p}{12} u(\text{AA}_3) + \dfrac{k_s - k_p}{12} u(\text{A}_1) & \text{SLIT, Asthma} \end{cases} \tag{31}
$$

for $t$ = 0, 1, 2, . . ., $T$, where:

- $k_s$ is the length of the grass-pollen season expressed in months (in the base case $k_s$ = 2, that is 60 days).

- $k_p$ (with $k_p \leq k_s$) is that part of the pollen season during which immunotherapy is more effective in symptom control than symptomatic therapy. When $k_p = k_s$ then AIT is more effective during the whole pollen season, and both clinical manifestations of ARC and perceived quality of life improve. When $k_p < k_s$ the clinical manifestations and quality of life improve only at the peak of the pollen season, whereas for the rest of the pollen season the QALY gained are identical to those gained by means of first-line symptomatic therapy.

In the base case, for both premature discontinuations or after the end of therapy, we assumed that the QALY gained were at the same level as those calculated with Formulas (30)

**Table 3. QALY gained in the base case in each cycle of Markov simulation, for each health state of the `SCIT` strategy.**

| t | 0 | 1 | 2 | 3 | 4 | 5 | 6 | 7 | 8 | 9 |
|---|---|---|---|---|---|---|---|---|---|---|
| ANA₂ | 0.96805 | 0.96805 | 0.96805 | 0.96805 | 0.96805 | 0.96805 | 0.96805 | 0.96805 | 0.96805 | 0.96805 |
| AA₂ | 0.59640 | 0.59640 | 0.59640 | 0.59640 | 0.59640 | 0.59640 | 0.59640 | 0.59640 | 0.59640 | 0.59640 |
| NANA₂ | 0.96805 | 0.96805 | 0.96805 | 0.96805 | 0.96805 | 0.96805 | 0.96805 | 0.96805 | 0.96805 | 0.96805 |
| NAA₂ | 0.59640 | 0.59640 | 0.59640 | 0.59640 | 0.59640 | 0.59640 | 0.59640 | 0.59640 | 0.59640 | 0.59640 |

and (31). For example, the evolution of QALY gained for patients undergone `SCIT` is shown in Table 3 (assuming a follow-up of $T = 9$ years):

These assumptions imply that the clinical effectiveness of immunotherapy is maintained for the entire duration of the follow-up. Moreover, the increased efficacy (with respect to symptomatic therapy) is already present from the first year of immunotherapy, and the protective effect is preserved for the entire follow-up even in case of premature discontinuation. These hypotheses are clearly unrealistic, and we are unable to understand whether the same set of assumption has been used by published pharmacoeconomic studies since, as stated before, most of these studies are not transparent about the calculation procedures that were actually used.

Expected QALYs for each cycle were calculated by summing the QALYs for each state by the proportion of the cohort in that state, and then adding these expected cycle QALYs across all cycles to obtain the total expected QALYs over the follow-up horizon. Similarly, expected costs for each cycle were calculated as a weighted averaged of costs pertaining to each state by the proportion of the cohort, with the particularity that patients prematurely abandoning the immunotherapy in a given cycle (either `SCIT` or `SLIT`) were charged of 50% of costs for immunotherapy in that cycle. Moreover, in the base case, indirect costs of lost productivity and transport cost were not charged to patients undergoing `SCIT`. The resulting total cost were calculated adding the expected costs across all cycles. All the expected values (QALYs and costs) were discounted at the annual rate $r = 0.035$ (3.5%). Traces of Markov cohort simulation of each health state in the base case, as well as of expected costs and QALYs can be appreciated in Fig 6.

Using the `no AIT` strategy as a comparator, the total expected costs and QALYs can be used to calculate an incremental cost-effectiveness ratio (ICER), that is the incremental cost for 1 additional QALY gained required for implementing the immunotherapy for reducing symptoms and asthma co-occurrence in patients suffering from grass-pollen seasonal ARC:

$$\text{ICER} = \frac{C_{\text{AIT}} - C_{\text{No AIT}}}{Q_{\text{AIT}} - Q_{\text{No AIT}}} = \frac{\Delta C}{\Delta Q} \tag{32}$$

where $C_{(\bullet)}$ indicates the total expected cost across the follow-up horizon (`AIT` = either `SCIT` or `SLIT`), while $Q_{(\bullet)}$ indicates the total expected QALYs gained by each of the three strategies. Estimate (32) is subject to uncertainty, that reflects the uncertainty in the model input parameters [67–69]. To demonstrate the possibility of an intervention being cost-effective at a certain willingness-to-pay (WTP) threshold, a probabilistic sensitivity analysis (PSA) is a valuable option, in which a prior sampling distribution of the model input variables is set [70, 71]. In the final section, we will briefly discuss on how our richly parameterized CEA model is well suited for a PSA. However, in this paper we will limit ourselves to the analysis of some selected scenarios, obtained by discontinuously varying one or more input parameters, having a two-fold objective in mind. Firstly, we can test whether our conclusions are robust under different circumstances, providing an early level sensitivity analysis. Secondly, these scenarios reflect

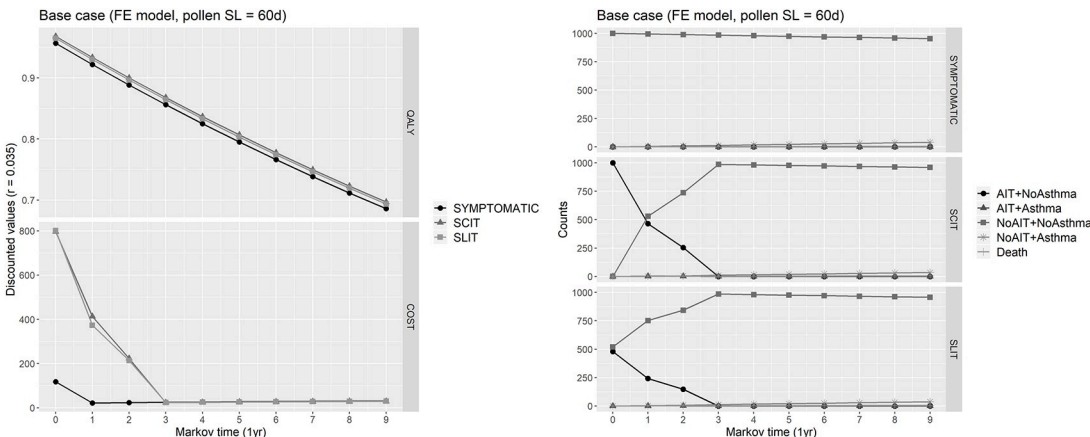

**Fig 6. Base case.** Right: Markov cohort evolution in the base case, where the fixed effects model has been used for calculating health states utilities, and for a pollen season length of 60 days. Left: Markov cohort evolution of expected costs and QALY gained for each Markov cycle.

situations that have a greater adherence to real-life and mirror plausible clinical assumptions, departing from the unrealistic assumptions underlying the base case. Specifically, we simulated the Markovian model under the following alternative settings:

- Base case with random effects (RE) model: this scenario is identical to the standard base case, except that the effects of interventions are meta-analyzed under the random effect model.

- No clinical benefit after premature discontinuation: this scenario is the most adverse to immunotherapy, as it is assumed that the higher level of QALYs provided by the administration of `SCIT` and `SLIT` is warranted only after completing a full cycle of three years of therapy (i.e. from $t = 3$ onwards we use Formulas (30) and (31)), while in $t = 0, 1, 2$ the utilities are calculated according to Formula (29) (valid for `no AIT`). Furthermore, for all the patients who prematurely discontinue therapy (i.e. before completing a full cycle of three years), the clinical benefit remains at the baseline level warranted by a purely symptomatic therapy (which is never discontinued), and QALYs are calculated with Formula 29 for each cycle.

- No asthma prevention by `AIT`: the only substantial difference with the base case is that `AIT` is assumed to not be more effective than a symptomatic therapy in preventing progression of ARC to asthma. This assumption corresponds to setting:

$$RR_{a(AIT)} = 1$$

Transition matrices must be modified accordingly. In particular, expression 15 becomes:

$$\text{pr}(\text{No AIT} + \text{No Asthma} \rightarrow \text{No AIT} + \text{Asthma}) =$$
$$= 0.76\, p_{a(1yr)} + 0.24\, p_{a(1yr)} = p_{a(1yr)} \approx 0.00463$$

with an identical change holding in expression (20) valid for `SCIT`.

- Full societal perspective: in this scenario, unlike the base case, the SCIT immunotherapy generates the following additional costs:

- Indirect non-healthcare costs due to the temporary interruption of work for administering therapy.

- Direct transportation costs.

For each of the above defined scenarios (included the base cases) we considered four different variation obtained in the following way:

- length of the follow-up $T = 9$, length of the grass-pollen season $S = 60$, $k_p = 2$ (the efficacy of the immunotherapy covers $P = k_p{}^*30 = 60$ days, the whole duration of the grass-pollen season).

- length of the follow-up $T = 9$, length of the grass-pollen season $S = 120$, $k_p = 2$.

- length of the follow-up $T = 14$, length of the grass-pollen season $S = 60$, $k_p = 2$.

- length of the follow-up $T = 9$, length of the grass-pollen season $S = 60$, $k_p = 0.5$ (the efficacy of the immunotherapy covers $P = 0.5{}^*30 = 15$ days, the peak of the grass-pollen season [61]).

Results are reported in Table 4, showing expected incremental costs, QALYs and the corresponding ICERs for each scenario and each possible variation. A graphical summary of the same results in the cost-effectiveness plane is shown in Fig 7. In synthesis, the average incremental QALYs across scenarios are 0.1452 (sd: 0.1252) for `SCIT` and 0.0767 (sd: 0.0516) for `SLIT`. In the same way, the average incremental costs across scenarios are €1420.542 (sd: €271.115) for `SCIT` and €1208.275 (sd: €15.695) for `SLIT`. It emerges that `SCIT` strategy offers, on average, a greater clinical benefit while being costlier than `SLIT` (as a consequence of the greater adherence to therapy, which is reflected in a greater amount of costs for immunotherapy). However, incremental costs of `SCIT` have a larger variability, because of indirect costs of lost productivity and transport cost that are a burden for the full societal scenario.

As far as ICERs are considered, a large variability in outcomes is present. We can observe that:

- `SCIT` systematically outperforms `SLIT`, except for the full societal perspective (for example, for $T = 9$ and a pollen season of 60 days, we have €16.729 for `SCIT` vs. €15,116 for `SLIT`).

- For longer pollen seasons (120 days) or longer follow-up duration ($T = 14$) the ICER decreases, because each patient experiences a greater clinical benefit over a larger time span, and QALYs gained per cycle increase accordingly.

- Assuming that no clinical benefit is achieved after premature discontinuation, and that at least three years of immunotherapy are required to improve clinical manifestations and perceiving a better quality of life, ICERs becomes far greater than €30,000. For example, for $T = 9$ and a pollen season of 60 days, we have €43,166 for `SCIT` €92,891 for `SLIT`.

- If the immunotherapy is effective only at the peak of the pollen season (15 days), the relative ICERs rise sharply. For example, when no clinical benefit is present after premature discontinuation, we have €74,770 for `SCIT` vs. €152,110 for `SLIT`.

- The distance between `SCIT` and `SLIT` in the base case increases when the interventions are meta-analyzed under the random effect model. This is an obvious consequence of the fact that the average back-transformed RTSS for `SCIT` is 2.5876 under the RE model vs. 0.8768 under the RE model.

Even in the light of these consideration we cannot conclude that both `SCIT` and `SLIT` (or only one of these two) could be considered cost-effective for ARC, as a reliable threshold value for cost-effectiveness is missing. For example, the British National Institute for Health and

**Table 4. Expected incremental costs (ΔC), expected incremental QALYs (ΔQ) and the relative ICERs (in 2018 process) for each main scenario and each possible variation (20 scenarios in total).** The length of the grass-pollen season $S$, and of the part $P$ of the pollen season during which immunotherapy is more effective in symptom control than symptomatic therapy, are expressed in days (d).

| Main scenario | Model | $T + 1$ | $S$ (d) | $P$ (d) | Strategy | ΔC | ΔQ | ICER (€) |
|---|---|---|---|---|---|---|---|---|
| Base case | FE | 10 | 60 | 60 | SCIT | 1256.1425 | 0.1139 | 11,028 |
| | | 10 | 60 | 60 | SLIT | 1216.8781 | 0.0805 | 15,116 |
| | | 10 | 120 | 120 | SCIT | 1389.2147 | 0.2086 | 6,660 |
| | | 10 | 120 | 120 | SLIT | 1181.5926 | 0.1486 | 7,951 |
| | | 15 | 60 | 60 | SCIT | 1241.2519 | 0.1675 | 7,410 |
| | | 15 | 60 | 60 | SLIT | 1208.2757 | 0.1178 | 10,257 |
| | | 10 | 60 | 15 | SCIT | 1256.1425 | 0.042 | 29,908 |
| | | 10 | 60 | 15 | SLIT | 1216.8781 | 0.0289 | 42,107 |
| Base case | RE | 10 | 60 | 60 | SCIT | 1256.1425 | 0.2642 | 4,755 |
| | | 10 | 60 | 60 | SLIT | 1216.8781 | 0.0804 | 15,135 |
| | | 10 | 120 | 120 | SCIT | 1389.2147 | 0.4989 | 2,785 |
| | | 10 | 120 | 120 | SLIT | 1181.5926 | 0.1483 | 7,968 |
| | | 15 | 60 | 60 | SCIT | 1241.2519 | 0.3810 | 3,258 |
| | | 15 | 60 | 60 | SLIT | 1208.2757 | 0.1176 | 10,274 |
| | | 10 | 60 | 15 | SCIT | 1256.1425 | 0.0872 | 14,405 |
| | | 10 | 60 | 15 | SLIT | 1216.8781 | 0.0289 | 42,107 |
| Premature discont. | FE | 10 | 60 | 60 | SCIT | 1256.1425 | 0.0291 | 43,166 |
| | | 10 | 60 | 60 | SLIT | 1216.8781 | 0.0131 | 92,891 |
| | | 10 | 120 | 120 | SCIT | 1389.2147 | 0.0444 | 31,289 |
| | | 10 | 120 | 120 | SLIT | 1181.5926 | 0.0192 | 61,541 |
| | | 15 | 60 | 60 | SCIT | 1241.2519 | 0.0507 | 24,482 |
| | | 15 | 60 | 60 | SLIT | 1208.2757 | 0.0229 | 52,763 |
| | | 10 | 60 | 15 | SCIT | 1256.1425 | 0.0168 | 74,770 |
| | | 10 | 60 | 15 | SLIT | 1216.8781 | 0.0080 | 152,110 |
| No asthma prevention | FE | 10 | 60 | 60 | SCIT | 1278.4711 | 0.0992 | 12,888 |
| | | 10 | 60 | 60 | SLIT | 1227.4419 | 0.0731 | 16,791 |
| | | 10 | 120 | 120 | SCIT | 1411.5424 | 0.1951 | 7,235 |
| | | 10 | 120 | 120 | SLIT | 1192.1564 | 0.1419 | 8,401 |
| | | 15 | 60 | 60 | SCIT | 1274.0865 | 0.1461 | 8,721 |
| | | 15 | 60 | 60 | SLIT | 1223.9551 | 0.1071 | 1,1428 |
| | | 10 | 60 | 15 | SCIT | 1278.4711 | 0.0276 | 46,321 |
| | | 10 | 60 | 15 | SLIT | 1227.4419 | 0.0217 | 56,564 |
| Full societal persp. | FE | 10 | 60 | 60 | SCIT | 1905.4595 | 0.1139 | 16,729 |
| | | 10 | 60 | 60 | SLIT | 1216.8781 | 0.0805 | 15,116 |
| | | 10 | 120 | 120 | SCIT | 2038.5317 | 0.2086 | 9,772 |
| | | 10 | 120 | 120 | SLIT | 1181.5926 | 0.1486 | 7,951 |
| | | 15 | 60 | 60 | SCIT | 1890.5689 | 0.1675 | 11,287 |
| | | 15 | 60 | 60 | SLIT | 1208.2757 | 0.1178 | 10,257 |
| | | 10 | 60 | 15 | SCIT | 1905.4595 | 0.0420 | 45,368 |
| | | 10 | 60 | 15 | SLIT | 1216.8781 | 0.0289 | 42,107 |

Care Excellence (NICE) considers cost-effective each technology whose ICER is not exceeding £30,000 per QALY gained [72]. However, other regulatory agencies adopt different thresholds. In the Netherlands for example, the threshold for a drug to be considered cost-effective varies from €10,000 to €80,000 depending on the severity of the disease.

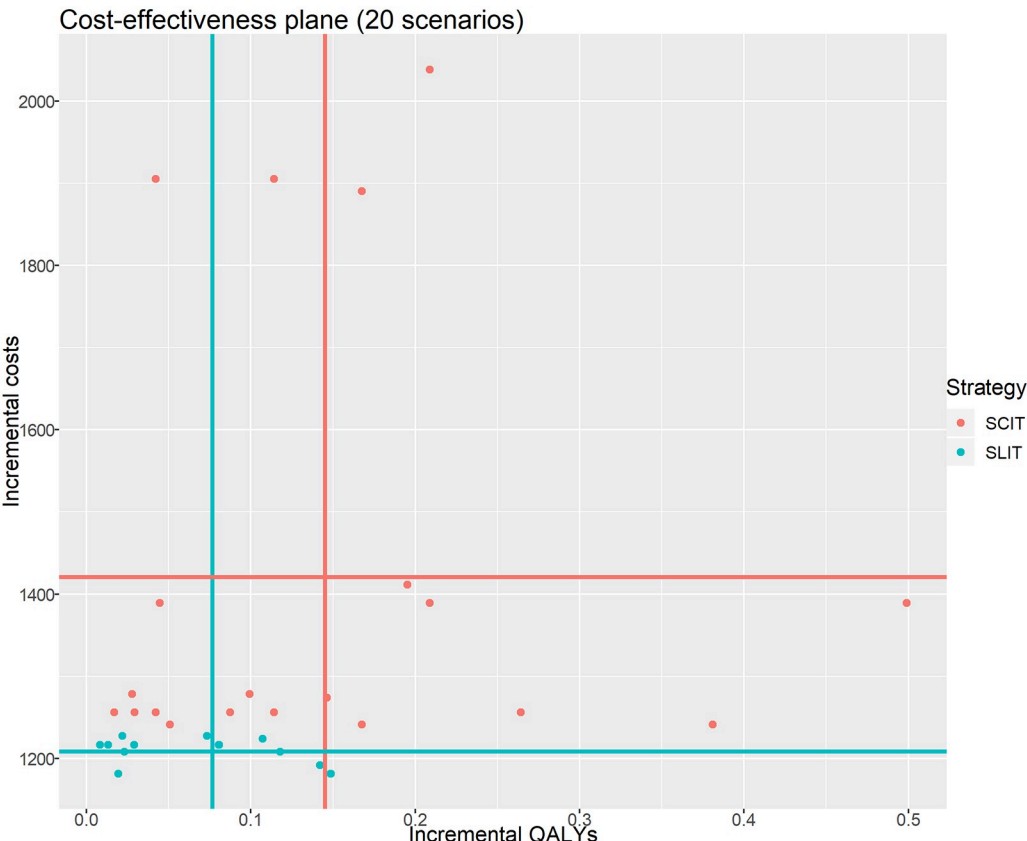

**Fig 7. Cost-effectiveness plane.** Expected incremental costs (ΔC, in 2018 prices) and expected incremental QALYs (ΔQ), for each scenario and each possible variation (20 scenarios in total).

## Discussion and conclusions

In this paper we have introduced a flexible, non-stationary Markov model for CEA of grass-pollen induced seasonal ARC. The model is intended to provide healthcare professionals and policymakers with useful and realistic information for cost-effective interventions. However, there are some limitations that are worth to be discussed further.

As mentioned earlier, the impact of model input parameters uncertainty on the reliability of our conclusions need to be investigated further. Indeed, in most of previously published studies, each input parameter is assigned a point estimate value together with a level of confidence around such estimate. In this way, a prior probability distribution can be assigned to all parameters in the model, and a Monte Carlo simulation to generate the sampling distribution of the joint mean cost and efficacy conducted, so that a quantification of the uncertainty surrounding those estimates can be obtained [73]. As already discussed, PSA has the advantage of indicating the probability of a health technology being cost-effective at various thresholds.

In our model, every clinical aspect can be linked to a specific input parameter. For example, it is quite immediate to introduce a prior probability distribution on the fraction of the pollen season during which the immunotherapy is effective, unifying and extending many scenarios that we have presented in Table 4. As $k_p$ is a fraction constrained in a given range (between 0 and $k_s$), the generalized Beta distribution provides a flexible way to specify a probability distribution over a particular range (with a specified minimum and maximum) [74]. Prior

distributions over the remaining input parameters of the model, such as log-relative risks and costs, are very commonly used in previous literature and hence will not be discussed further [28]. On the contrary, it is much more difficult to program and compute a probabilistic Monte Carlo simulation that includes, as a particular case, all the scenarios we have analyzed before. For example, utilities of the scenario in which no clinical benefit is present if the immunotherapy is prematurely discontinued, are calculated differently than the base case (Formulas (29), (30) and (31)). In other words, we are facing a discontinuity between the two scenarios that cannot fit in a prior probability distribution. In this case, we could experiment Bernoulli priors to randomly switch between the two scenarios inside the same probabilistic simulation. Future papers might fruitfully further explore these approaches and issues.

In conclusion, our model has demonstrated the ability to study the cost-effectiveness of the immunotherapy for ARC, simulating scenarios that are typical of real-life clinical practice and that cannot fall within the hypotheses underlying RCTs, whose conclusions are sometimes unrealistic and generate ICER estimates that are biased in favor of the cost-effectiveness of interventions. However, as we have already pointed in earlier sections, many are the processes to be refined to make the Markov simulation model even more realistic, and to incorporate the uncertainty about model input parameters in a generalizable way. Indeed, these are interesting topics for future researches, as more investigations are necessary to validate the conclusions that could be drawn from this current study.

## Author Contributions

**Conceptualization:** Massimo Bilancia, Danilo Di Bona.

**Data curation:** Massimo Bilancia.

**Investigation:** Giuseppe Pasculli.

**Methodology:** Massimo Bilancia.

**Software:** Massimo Bilancia.

**Supervision:** Massimo Bilancia, Danilo Di Bona.

**Writing – original draft:** Massimo Bilancia, Giuseppe Pasculli, Danilo Di Bona.

**Writing – review & editing:** Massimo Bilancia, Giuseppe Pasculli, Danilo Di Bona.

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
