## [Decision Letter · Decision Letter 0]

8 Jan 2020

PONE-D-19-29347

A non-stationary Markov model for economic evaluation of grass pollen allergoid immunotherapy

PLOS ONE

Dear Prof. BILANCIA,

Thank you for submitting your manuscript to PLOS ONE. After careful consideration, we feel that it has merit but does not fully meet PLOS ONE’s publication criteria as it currently stands. Therefore, we invite you to submit a revised version of the manuscript that addresses the points raised during the review process.

Although both the reviewers assessed that the paper is technically sound and contributes to the economic evaluation literature pertaining to grass pollen allergoid immunotherapy, reviewer 2 pointed to many grammatical and typographical errors, had reservations in terms of the way the narratives are framed, and assessed that many revisions must be made in order to effectively and clearly convey the science both in-text and in the figures/tables. The paper needs to be revised by addressing each point by referee 2 and preparing a 'Author response to reviewer comments' document. This is a long paper; there are scopes for making it more concise.

Additionally, the manuscript requires major revisions in terms of content presentation and narratives. For instance, the abstract headings could follow typical style of Introduction, methods, results, conclusion (without the use of bullets); the in-text cited references should be presented in order (e.g. in page 3 the authors used the references (22, 44, 54) just after the in-text citation of reference 15,16).

The authors must revise the manuscript by addressing comments from the reviewers and prepare a 'Author response to reviewer comments' document. Also, the paper needs to be copy-edited for typos and in compliance to the journal styles.

We would appreciate receiving your revised manuscript by Feb 22 2020 11:59PM. To enhance the reproducibility of your results, we recommend that if applicable you deposit your laboratory protocols in protocols.io, where a protocol can be assigned its own identifier (DOI) such that it can be cited independently in the future. For instructions see: http://journals.plos.org/plosone/s/submission-guidelines#loc-laboratory-protocols

We look forward to receiving your revised manuscript.

Kind regards,

Muhammad Jami Husain, Ph.D.

Academic Editor

PLOS ONE

2. We note you have included a table to which you do not refer in the text of your manuscript. Please ensure that you refer to Table 6 in your text; if accepted, production will need this reference to link the reader to the Table.

Reviewers' comments:

Reviewer's Responses to Questions

**Comments to the Author**

1. Is the manuscript technically sound, and do the data support the conclusions?

Reviewer #1: Yes

Reviewer #2: Yes

2. Has the statistical analysis been performed appropriately and rigorously? 

Reviewer #1: Yes

Reviewer #2: Yes

3. Have the authors made all data underlying the findings in their manuscript fully available?

Reviewer #1: Yes

Reviewer #2: Yes

4. Is the manuscript presented in an intelligible fashion and written in standard English?

Reviewer #1: Yes

Reviewer #2: Yes

5. Review Comments to the Author

Reviewer #1: The manuscript is well written. The authors developed a flexible, non-stationary Markovian model for cost-effective analysis (CEA) of grass-pollen induced seasonal ARC. The goal of developing such model is to provide healthcare professionals and policymakers with useful and realistic information for cost-effective interventions. Although there is no a specific section of conclusion, the purpose of the work has been presented clearly throughout the whole manuscript. It will be more convincing if the authors can apply the non-stationary Markovian model to a set of real data collected from a local or national hospital patients.

Reviewer #2: This paper presents novel scientific evidence that could be useful for healthcare professionals/practitioners and policymakers concerning grass pollen allergoid immunotherapy. The authors have a very strong background in statistics, health economics, and especialy cost-effectiveness analysis. Robust methods are used and strong assumptions supported by the literature are used for the models. However, revisions must be made in order to effectively and clearly convey the science both in-text and in the figures/tables. The following are requirements and suggestions for revisions:

*Throughout the paper, minimize the use of adding information into parentheses. It is acceptable to do this sometimes; however, it is done too frequently. Removing the parentheses and re-writing the sentences make it more scientific. The use of so many parentheses makes the tone more casual, as if the authors were trying to interject thoughts, rather than stating necessary information.

*For all of the tables, include footnotes on abbreviations and any other relevant and necessary information.

-Lines 46 and 50-53: The use of so many parentheses are distracting and read as run-in sentences. Re-write these sentences for better clarity and readibility.

-Line #53: Using words and phrases such as "it is important to remember" change the tone of this scientific paper to a more conversational tone and infers a some subjectivity that is more appropriate for the Discussion. Suggest removing this phrase.

-Line #54: Remove "that".

-Line #61: Remove "it".

-Line #67: The use of "indeed" is used too frequently.

-Line #74: Add a comma after "increased".

-Lines #75-#78: Ambiguous and lengthy sentence; re-word.

-Line #79: Change "an" to "a once-daily..."

-Line #90: Consider starting a new paragraph with "However...".

-Line #90: As mentioned previously, phrases that contain "more importantly" infer a sense of subjectivity - this tone and language is more appropriate in the Discussion section. Suggest removing "even more importantly".

-Lines #93-95: I have two questions for this sentence: "For example, [58] estimated that total cost of the ARC condition in the USA, in 1994 Dollars, was $1.23 billion (95% confidence interval, $846 million to $1.62 billion), with direct medical 95 expenses accounting for 94% of total costs." (1) The authors include a reference number [58] but do not state who estimated the total cost of the ARC condition in the USA - Who conducted the estimation/What is the source of this information? (2) The authors reference a study that used 1994 USD. These dollar amounts are quite outdated, since 1994 is now around 26 years ago. Is there a more recent similar study that has more current estimates?

-Line #97: Again the authors include a reference [59] but fail to mention who esimated that total economic burden in Italy.

-Lines #105-#109: Run-on sentence; revise.

-Line #111: Reference [54] stated but who or what study found...? Please fix all subsequent similar instances throughout the paper.

-Lines #111-#115: Run-on sentence; revise.

-Line #121: This sentence needs to be revised. Again, there is a sense of a more casual tone wih the first two clauses. Remove "we decided". Simply state what you did, such as "we developed" and "we describe." In addition, it is also too lengthy and is a run-on sentence. Throughout the paper, it is imperative to vary sentence lenghth; however, many sentences need to be shortened to increase clarity and readability.

-Line #128: Simply stating, "which help to..." is too bold of a conclusion. Revise to say something such as, "which may inform decision-making on selecting the option with the best value."

-Line #130: Change "will be focused" to "focuses on"

-Line #132: "However" is used too frequently thoughout. Use different transitions throughout the paper. Change "we will also" to "we discuss" Remove "some"

-Lines #132-136: Run-on sentence.

-Line #155: Define "mutatis mutandis"; a footnote would be acceptable.

-Line #159: Remove "obvious".

-Line #165: Again, missing information [21].

-Line #168: The author did not include a hyphen earlier for "comorbidity." Different countries have different conventions concerning the hyphen, so be consistent. If the author uses a hyphen, like in this line for "co-existing" then the authors should be consistent throughout.

-Line #179: Re-write to "Although this is not mandatory, a longer follow-up might be clinically justifiable."

-Line #185: Remove the comma after "year".

-Lines #187-189: Is there evidence in the literature that can support the assumption made from the statement: "[We] assumed that asthmatic disease could not complicate ARC already from the first year."

-Line #190: Provide a reference for the claim, "the therapy is administered for a maximum of three years, in combination with symptomatic therapy."

-Line #211: Remove "Needless to say".

-Line #242: In the paragraph, either spell out "page" or abbreviate as "pg".

-Line #243: Add "the" in front of "Italian National Institute of Statistics (ISTAT)".

-Line #258: Remove "that"; past tense also changes here suddently. In this section, decide on which verb tense you want and stick with it for consistency.

-Lines #262-266: Run-on and ambiguous. Re-word for better clarity.

-Line #264: Remove the "s" to make it "treatment".

-Line #280: Change the comma to a semi-colon and remove "and".

-Line #292: Remove "an".

-Lines #287-290: Run-on; revise.

-Line #296: Change "loose" to "lose".

-Line #301: Remove "a" before "reaching a state..."

-Line #328: Remove "Of course".

-Line #388: Remove "have" in front of "considered".

*Table 2 seems too small to be a standalone table. I suggest referring to it as a figure.

-Line #458: Remove "decided" and simply state what you did. Please do this for other instances throughout the paper. For example, re-write the sentence to "We did not consider intangible costs..."

-Line #498: Add a space in-between "200" and "mcg".

-Lines #504-539: Be consistent with periods - either include them or not.

-Line #570: Remove "costs" after "GRAZAZ".

-Lines #552-561: Make sure to close the parentheses.

-Line #562: Remove "we" and change to "As 2 packs are needed to administer 10 injections..."

-Lines #463-603 (including Table 5). Consider combining Tables 4 and 5. Furthermore, I highly suggest to pull all of the information (direct and indirect costs) from the entire section ("Cost") into one consolidated figure and name it something along the lines of "Direct and Indirect Costs.."

-Line #654: Remove "a" before "prior" or change "distributions" to "distribution".

*Table 7: Consider re-formatting the table so you do not have to repeat the same information in each line.

*There are many diagrams included. Consider choosing which ones to keep in the body of the manuscript or include in an Appendix.

6. PLOS authors have the option to publish the peer review history of their article (what does this mean?). If published, this will include your full peer review and any attached files.

Reviewer #1: No

Reviewer #2: No

---

## [Author Response · Author response to Decision Letter 0]

15 Feb 2020

To the Academic Editor

Muhammad Jami Hussain, Ph.D.

Thank for your kind comments. We have tried to respond appropriately to each point addressed by the two anonymous referees. The paper has been accurately edited for typos, and it has also been revised by an English mother-tongue reader. We have carefully checked that our manuscript meets PLOS ONE's style requirements and naming conventions. With regard to the points you specifically mentioned:

1) we have carefully revised the references and the ordering of the in-text numbered citations, which have been arranged by order of appearance in the revised version of the paper. We have also corrected two citations which were wrongly formatted according to the APA citation style. 

2) the abstract has been rewritten following a more narrative style, and abstract headings are now arranged according to a typical subdivision into Introduction, Methods, Results and Conclusions. The use of bullets, which has a twofold purpose, has been motivated by the true nature of the paper itself. On one side, our intention is to provide novel scientific evidence to healthcare professionals, conveying useful information aimed at cost-effective interventions for allergic rhino-conjunctivitis (ARC). On the other side, we are also interested in the algorithmic aspects of our proposal. Pharmacoeconomics models are Markov models on a graph characterized by many input parameters, which leave room for arbitrary choices and lack of robustness. And in fact, many published papers are affected by these kinds of issues (see, for example, reference [23] for an interesting systematic review of economic evaluations of SCIT and SLIT therapies in adults). For these reasons, we described in detail all the structural components of our model, as well as its input parameters, in order to facilitate the reproducibility and the circulation of its contents. Therefore in summary, the 'hybrid' nature of our paper led to using bullets as a fast and concise way to convey details about input parameters, utilities and costs in an unambiguous manner. A more narrative style would likely prove unsuitable, maybe resulting in an longer paper as well. 

3) in the original version of the paper the reference to Table 6 was contained in the sentence 'in the Table below'. In the current version we directly refer to Table 6 in the text, using a Latex \\ref{…} command to the corresponding \\label{…} command.

Reviewer #1

Thank you for your kind observations and comments. With regards to questions you specifically raised:

The manuscript is well written. The authors developed a flexible, non-stationary Markovian model for cost-effective analysis (CEA) of grass-pollen induced seasonal ARC. The goal of developing such model is to provide healthcare professionals and policymakers with useful and realistic information for cost-effective interventions. Although there is no a specific section of conclusion, the purpose of the work has been presented clearly throughout the whole manuscript. It will be more convincing if the authors can apply the non-stationary Markovian model to a set of real data collected from a local or national hospital patients.

Thank for your interesting comment. Our model simulates a cohort of individuals over a given timespan, and it is thus unsuitable for your proposed application. Indeed, the model is not estimated from data, but it is simulated using a collection of input parameters which should capture all the relevant details in a realistic way: the structure of the population at risk, the epidemiological aspects, rates of treatment discontinuation and probabilities of progression towards more severe states (such as asthma), utilities, costs, etc… All of this means that your question is perfectly sensible, and can be reformulated as: what is the impact of model input parameters on the reliability of our conclusions? The correct way to answer this question is to assign a prior probability distribution to all parameters in the model, and to generate via Monte Carlo simulation the joint sampling distribution of both costs and benefits. By means of this probabilistic sensitivity analysis (PSA) we will then be able to estimate the probability of being cost-effective at a certain willingness-to-pay (WTP) threshold. The latter WTP threshold can vary substantially as a function of subgroups of patients, as well as of country-specific parameters. These aspects were all discussed in detail in the Discussion and Conclusion section, where we state that PSA is a valuable option to make our conclusion more robust. However, implementation of PSA requires specifying an expanded probabilistic model and therefore a separate full paper. A thorough answer to your question is object of ongoing research.

Reviewer #2

Thank for your clear and precise revision. We have carefully implemented most of the suggested changes; in particular, the use of parenthesis has been kept to a minimum throughout the revised text, and the paper has been post-edited and revised by an English mother-tongue reader. Hereinafter, we respond in detail to the most relevant points you raised:

-Lines #93-95: I have two questions for this sentence: "For example, [58] estimated that total cost of the ARC condition in the USA, in 1994 Dollars, was $1.23 billion (95% confidence interval, $846 million to $1.62 billion), with direct medical 95 expenses accounting for 94% of total costs." (1) The authors include a reference number [58] but do not state who estimated the total cost of the ARC condition in the USA - Who conducted the estimation/What is the source of this information? (2) The authors reference a study that used 1994 USD. These dollar amounts are quite outdated, since 1994 is now around 26 years ago. Is there a more recent similar study that has more current estimates?

We have specified that the data source is the National Medical Expenditure Survey; the paper by Malone et al. (1997) is somewhat outdated, but it is historically significant because it was the first realistic estimate of cost of illness of seasonal allergic rhino-conjunctivitis (ARC). We have not used any estimates from this paper in our simulation. Immediately following, we have reported less outdated evidence provided by a paper in an EU context, where costs are expressed in 2015 Euros, fully confirming the economic burden associated with cost of illness of ARC.

-Line #155: Define "mutatis mutandis"; a footnote would be acceptable.

Footnotes are forbidden in the PLOS ONE's style requirements. We replaced "mutatis mutandis" with "after the necessary changes have been made".

-Lines #187-189: Is there evidence in the literature that can support the assumption made from the statement: "[We] assumed that asthmatic disease could not complicate ARC already from the first year."

-Line #190: Provide a reference for the claim, "the therapy is administered for a maximum of three years, in combination with symptomatic therapy."

These assumptions are not evidence-based, but rather have often the function to keep the R code simpler, without any loss of generality of the conclusions reached. For example, under the assumption that asthmatic disease could not complicate ARC already from the first year, the initial state vectors takes a particularly simple form. What matters most is that the same assumption is valid for all the three scenarios, and thus no scenario is under- or over-weighted relative to others. Put it another way, removing this assumption would not alter the relative distance between costs and benefits across the various scenarios. 

The assumption that the therapy is administered for a maximum of three years has a similar meaning. The literature recommends that SCIT or SLIT should be continued for at least 3 years (see references [19,20] in the revised version). If we had assumed that immunotherapy was administered, for example, over a six-year timeframe, we would have reached identical conclusions (except for irrelevant changes in the ICERs) provided that the same assumption had been used for both SCIT and SLIT. In our case, we opted for a three-year timeframe because we had reliable data on immunotherapy discontinuation up to the 3rd year of treatment. Besides, particularly in the case of SLIT, very few patients continue immunotherapy for more than three years.

The use of such "boundary conditions" is quite common in simulation-based cost-effectiveness studies, and it is perfectly legitimate provided that: a) the assumptions are not entirely unrealistic; b) they are not used for unduly over-weighting benefits or under-weighting costs of a single scenario.

*Table 2 seems too small to be a standalone table. I suggest referring to it as a figure.

The single-row Table 2 looked unpleasant. We reformatted it as a better-looking name-value-pair list.

-Lines #463-603 (including Table 5). Consider combining Tables 4 and 5. Furthermore, I highly suggest to pull all of the information (direct and indirect costs) from the entire section ("Cost") into one consolidated figure and name it something along the lines of "Direct and Indirect Costs.."

That is exactly what we wanted to avoid doing. Many cost-effectiveness published studies present tables of direct and indirect cost, as well as of base input parameters, but it is often not transparent neither the way in which these parameters have been calculated, nor how they are in turn used to calculate total costs and benefits. Many published papers are affected by these kinds of issues (see, for example, reference [23] for an interesting systematic review of economic evaluations of SCIT and SLIT therapies in adults). For these reasons, the very nature of our paper is twofold. On one side, our intention is to provide novel scientific evidence to healthcare professionals, conveying useful information aimed at cost-effective interventions for allergic rhino-conjunctivitis (ARC). On the other side, we are also interested in describing in detail all the structural components of our model, as well as its input parameters, in order to facilitate its reproducibility and circulation, and circumvent the above-mentioned transparency issues. Put in another words, we were more interested in providing detailed information on the calculation of costs and utilities, rather than providing a crude listing of these values.

*Table 7: Consider re-formatting the table so you do not have to repeat the same information in each line.

We do not understand how the table could be re-formatted. Each line is a unique combination of scenario, meta-analysis estimation model, number of cycles, length of the grass-pollen season, length of the part of the pollen season during which immunotherapy is more effective in symptom control than symptomatic therapy, strategy. We cannot see a way in which all this information could be further compressed.

*There are many diagrams included. Consider choosing which ones to keep in the body of the manuscript or include in an Appendix.

This is a lengthy paper. Unfortunately, the proposed time-inhomogeneous Markov model is not very easy to understand without any graphical aid. Thus, we believe that all the seven diagrams included should be kept in the body of the manuscript.

---

## [Decision Letter · Decision Letter 1]

26 Feb 2020

PONE-D-19-29347R1

A non-stationary Markov model for economic evaluation of grass pollen allergoid immunotherapy

PLOS ONE

Dear Prof. BILANCIA,

Thank you for submitting your manuscript to PLOS ONE. After careful consideration, we feel that it has merit but does not fully meet PLOS ONE’s publication criteria as it currently stands. Therefore, we invite you to submit a revised version of the manuscript that addresses the points raised during the review process.

We would appreciate receiving your revised manuscript by Apr 11 2020 11:59PM. To enhance the reproducibility of your results, we recommend that if applicable you deposit your laboratory protocols in protocols.io, where a protocol can be assigned its own identifier (DOI) such that it can be cited independently in the future. For instructions see: http://journals.plos.org/plosone/s/submission-guidelines#loc-laboratory-protocols

We look forward to receiving your revised manuscript.

Kind regards,

Muhammad Jami Husain, Ph.D.

Academic Editor

PLOS ONE

Additional Editor Comments (if provided):

Thanks for revising the paper, addressing the reviewers' comments. We request the authors to take into account the remarks from the second reviewer on the revised submission; and submit a revised version.

Reviewers' comments:

Reviewer's Responses to Questions

**Comments to the Author**

1. If the authors have adequately addressed your comments raised in a previous round of review and you feel that this manuscript is now acceptable for publication, you may indicate that here to bypass the “Comments to the Author” section, enter your conflict of interest statement in the “Confidential to Editor” section, and submit your "Accept" recommendation.

Reviewer #2: All comments have been addressed

2. Is the manuscript technically sound, and do the data support the conclusions?

Reviewer #2: Yes

3. Has the statistical analysis been performed appropriately and rigorously? 

Reviewer #2: Yes

4. Have the authors made all data underlying the findings in their manuscript fully available?

Reviewer #2: Yes

5. Is the manuscript presented in an intelligible fashion and written in standard English?

Reviewer #2: Yes

6. Review Comments to the Author

Reviewer #2: Dear Authors,

This is a very much improved version of the manuscript. Thank you very much for the hard work that was put in. All comments are adequately answered and addressed. However, I have a few minor suggestions, in which I will leave the final decisions to the Editor. Thank you. Great job!

***BEGIN RESPONSES***

Reviewer #2

Thank for your clear and precise revision. We have carefully implemented most of the suggested changes; in particular, the use of parenthesis has been kept to a minimum throughout the revised text, and the paper has been post-edited and revised by an English mother-tongue reader. Hereinafter, we respond in detail to the most relevant points you raised.

***Thank you very much for making the changes that were required, to many of the suggestions, and providing clarifications and answers to my questions. This revised draft is very much improved and now effectively conveys the science in a clearer, more concise manner. Excellent work.

***There is, however, one sentence that needs to be revised in the abstract. The authors accidentally use the word, "experiment" instead of "experience." This sentence should be changed to, "For longer pollen seasons or longer follow-up duration, the ICER decreases because each patient experiences a greater clinical benefit over a larger time span, and quality-adjusted life years (QALYs) gained per cycle increase accordingly."

-Lines #93-95: I have two questions for this sentence: "For example, [58] estimated

that total cost of the ARC condition in the USA, in 1994 Dollars, was $1.23 billion (95% confidence interval, $846 million to $1.62 billion), with direct medical 95 expenses accounting for 94% of total costs." (1) The authors include a reference number [58] but do not state who estimated the total cost of the ARC condition in the USA - Who conducted the estimation/What is the source of this information? (2) The authors reference a study that used 1994 USD. These dollar amounts are quite outdated, since 1994 is now around 26 years ago. Is there a more recent similar study that has more current estimates?

We have specified that the data source is the National Medical Expenditure Survey; the

paper by Malone et al. (1997) is somewhat outdated, but it is historically significant

because it was the first realistic estimate of cost of illness of seasonal allergic rhinoconjunctivitis. (ARC). We have not used any estimates from this paper in our simulation. Immediately following, we have reported less outdated evidence provided by a paper in an EU context, where costs are expressed in 2015 Euros, fully confirming the economic burden associated with cost of illness of ARC.

***Thank you for adding this additional information. It answers my questions.

-Line #155: Define "mutatis mutandis"; a footnote would be acceptable.

Footnotes are forbidden in the PLOS ONE's style requirements. We replaced "mutatis

mutandis" with "after the necessary changes have been made".

***This is fine.

-Lines #187-189: Is there evidence in the literature that can support the assumption

made from the statement: "[We] assumed that asthmatic disease could not complicate

ARC already from the first year."

-Line #190: Provide a reference for the claim, "the therapy is administered for a

maximum of three years, in combination with symptomatic therapy."

These assumptions are not evidence-based, but rather have often the function to keep

the R code simpler, without any loss of generality of the conclusions reached. For

example, under the assumption that asthmatic disease could not complicate ARC

already from the first year, the initial state vectors takes a particularly simple form. What matters most is that the same assumption is valid for all the three scenarios, and thus no scenario is under- or over-weighted relative to others. Put it another way, removing this assumption would not alter the relative distance between costs and benefits across the various scenarios. The assumption that the therapy is administered for a maximum of three years has a similar meaning. The literature recommends that SCIT or SLIT should be continued for at least 3 years (see references [19,20] in the revised version). If we had assumed that immunotherapy was administered, for example, over a six-year timeframe, we would have reached identical conclusions (except for irrelevant changes in the ICERs) provided that the same assumption had been used for both SCIT and SLIT. In our

Powered by Editorial Manager® and ProduXion Manager® from Aries Systems Corporation

case, we opted for a three-year timeframe because we had reliable data on immunotherapy discontinuation up to the 3rd year of treatment. Besides, particularly in the case of SLIT, very few patients continue immunotherapy for more than three years. The use of such "boundary conditions" is quite common in simulation-based costeffectiveness studies, and it is perfectly legitimate provided that: a) the assumptions are not entirely unrealistic; b) they are not used for unduly over-weighting benefits or under-weighting costs of a single scenario.

***This explanation make sense. Thank you for clarifying.

*Table 2 seems too small to be a standalone table. I suggest referring to it as a figure.

The single-row Table 2 looked unpleasant. We reformatted it as a better-looking name value-pair list.

***This is fine.

-Lines #463-603 (including Table 5). Consider combining Tables 4 and 5. Furthermore,

I highly suggest to pull all of the information (direct and indirect costs) from the entire section ("Cost") into one consolidated figure and name it something along the lines of "Direct and Indirect Costs.."

That is exactly what we wanted to avoid doing. Many cost-effectiveness published

studies present tables of direct and indirect cost, as well as of base input parameters, but it is often not transparent neither the way in which these parameters have been calculated, nor how they are in turn used to calculate total costs and benefits. Many published papers are affected by these kinds of issues (see, for example, reference [23] for an interesting systematic review of economic evaluations of SCIT and SLIT therapies in adults). For these reasons, the very nature of our paper is twofold. On one side, our intention is to provide novel scientific evidence to healthcare professionals, conveying useful information aimed at cost-effective interventions for allergic rhinoconjunctivitis (ARC). On the other side, we are also interested in describing in detail all the structural components of our model, as well as its input parameters, in order to facilitate its reproducibility and circulation, and circumvent the above-mentioned transparency issues. Put in another words, we were more interested in providing detailed information on the calculation of costs and utilities, rather than providing a crude listing of these values.

***Thank you for your detailed explanations and reasonings behind not consolidating the figures/tables. However, I still feel like Tables 3 and 4 (formerly Tables 4 and 5) are too small to be stand alone. In addition, cutting down on these tables could help with some space, as there are many graphs and diagrams to be included, and the paper itself is very long, as mentioned by the authors and the Editor. Since there are no costs to be reported after Year 3 for both tables, I feel that the data can simply be reported in the body of the manuscript, in bullet-point format like what is already presented, to be consistent with the rest of the paper. For instance, bullet points (placed appropriately in the text) can describe the costs from Table 3: Years 0-2, and another bullet point can state that after Year 3, none of the strategies incurred any cost for medical resources. The same is suggested for the data in Table 4, indirect costs of lost productivity or costs of transport for SCIT. However, this is just a suggestion, and I will leave this up to the Editor.

*Table 7: Consider re-formatting the table so you do not have to repeat the same information in each line.

We do not understand how the table could be re-formatted. Each line is a unique combination of scenario, meta-analysis estimation model, number of cycles, length of the grass-pollen season, length of the part of the pollen season during which immunotherapy is more effective in symptom control than symptomatic therapy, strategy. We cannot see a way in which all this information could be further compressed.

***I see the differences now, that each line is unique. However, because the same words repeat so much, it was hard on the eyes upon first glace. Suggestion: For the "Main scenario," consider stating each scenario only once in the box. For instance, instead of repeating "Base Case" 8 times in the first box, simply state it once. Same suggestion for "Model". I will leave this up to the Editor.

*There are many diagrams included. Consider choosing which ones to keep in the body of the manuscript or include in an Appendix.

This is a lengthy paper. Unfortunately, the proposed time-inhomogeneous Markov model is not very easy to understand without any graphical aid. Thus, we believe that all the seven diagrams included should be kept in the body of the manuscript.

***I understand the rationale, but I will leave this decision up to the Editor.

7. PLOS authors have the option to publish the peer review history of their article (what does this mean?). If published, this will include your full peer review and any attached files.

Reviewer #2: No

---

## [Author Response · Author response to Decision Letter 1]

8 Apr 2020

To the Academic Editor

Muhammad Jami Hussain, Ph.D.

Dr Prof. Hussain,

as requested, we tried to take into account all the remarks from the second reviewer on the revised submission. Thank you for your valuable help and assistance.

Reviewer #2

We wanted to thank you for your help and suggestions, through which we were definitely able to improve the revised version of the draft. With respect to the new points you raised:

***There is, however, one sentence that needs to be revised in the abstract. The authors accidentally use the word, "experiment" instead of "experience." This sentence should be changed to, "For longer pollen seasons or longer follow-up duration, the ICER decreases because each patient experiences a greater clinical benefit over a larger time span, and quality-adjusted life years (QALYs) gained per cycle increase accordingly."

 Thank you for your suggestion. We have now corrected this mistake.

***Thank you for your detailed explanations and reasonings behind not consolidating the figures/tables. However, I still feel like Tables 3 and 4 (formerly Tables 4 and 5) are too small to be stand alone. In addition, cutting down on these tables could help with some space, as there are many graphs and diagrams to be included, and the paper itself is very long, as mentioned by the authors and the Editor. Since there are no costs to be reported after Year 3 for both tables, I feel that the data can simply be reported in the body of the manuscript, in bullet-point format like what is already presented, to be consistent with the rest of the paper. For instance, bullet points (placed appropriately in the text) can describe the costs from Table 3: Years 0-2, and another bullet point can state that after Year 3, none of the strategies incurred any cost for medical resources. The same is suggested for the data in Table 4, indirect costs of lost productivity or costs of transport for SCIT. However, this is just a suggestion, and I will leave this up to the Editor.

 After a careful reading, we fully accepted your suggestion. Tables 3 and 4 (formerly Tables 4 and 5) are now removed as they informed too little. Both tables were substituted with a more informative bulleted list, indicating the average monetary value for Years 0 - 2, and that none of the strategies incurred any additional cost for medical resources for Year 3 and afterwards. Thank you again for the useful advice.

*Table 7: Consider re-formatting the table so you do not have to repeat the same information in each line.

We do not understand how the table could be re-formatted. Each line is a unique combination of scenario, meta-analysis estimation model, number of cycles, length of the grass-pollen season, length of the part of the pollen season during which immunotherapy is more effective in symptom control than symptomatic therapy, strategy. We cannot see a way in which all this information could be further compressed.

***I see the differences now, that each line is unique. However, because the same words repeat so much, it was hard on the eyes upon first glace. Suggestion: For the "Main scenario," consider stating each scenario only once in the box. For instance, instead of repeating "Base Case" 8 times in the first box, simply state it once. Same suggestion for "Model". I will leave this up to the Editor.

 We have reformatted Table 4 (formerly Table 7) on the ground of your advice. The overall readability is now much more improved. Thank you.

---

## [Decision Letter · Decision Letter 2]

22 Apr 2020

A non-stationary Markov model for economic evaluation of grass pollen allergoid immunotherapy

PONE-D-19-29347R2

Dear Dr. BILANCIA,

We are pleased to inform you that your manuscript has been judged scientifically suitable for publication and will be formally accepted for publication once it complies with all outstanding technical requirements.

With kind regards,

Muhammad Jami Husain, Ph.D. 

Collection Editor

PLOS ONE

Additional Editor Comments (optional):

The reviewer went through the second revision and assessed that the authors have addressed all the comments. The paper may be accepted for publication; kindly requesting the authors to carefully review any proof-read/copy-editing issues, if any.

Reviewers' comments:

Reviewer's Responses to Questions

**Comments to the Author**

1. If the authors have adequately addressed your comments raised in a previous round of review and you feel that this manuscript is now acceptable for publication, you may indicate that here to bypass the “Comments to the Author” section, enter your conflict of interest statement in the “Confidential to Editor” section, and submit your "Accept" recommendation.

Reviewer #2: All comments have been addressed

2. Is the manuscript technically sound, and do the data support the conclusions?

Reviewer #2: Yes

3. Has the statistical analysis been performed appropriately and rigorously? 

Reviewer #2: Yes

4. Have the authors made all data underlying the findings in their manuscript fully available?

Reviewer #2: Yes

5. Is the manuscript presented in an intelligible fashion and written in standard English?

Reviewer #2: Yes

6. Review Comments to the Author

Reviewer #2: Dear Authors,

Thank you very much for making the suggested changes. The manuscript flows nicely in a scientific tone throughout, and the revisions that were made clearly convey and communicate the science. Excellent work.

7. PLOS authors have the option to publish the peer review history of their article (what does this mean?). If published, this will include your full peer review and any attached files.

Reviewer #2: No

---

## [Editor Report · Acceptance letter]

29 Apr 2020

PONE-D-19-29347R2 

A non-stationary Markov model for economic evaluation of grass pollen allergoid immunotherapy 

Dear Dr. Bilancia:

I am pleased to inform you that your manuscript has been deemed suitable for publication in PLOS ONE. Congratulations! Your manuscript is now with our production department. 

With kind regards,

on behalf of

Dr. Muhammad Jami Husain 

Collection Editor

PLOS ONE